# A mechanism of global gene expression regulation is disrupted by multiple disease states and drug treatments

Benjamin S. Pickard  *

Strathclyde Institute of Pharmacy and Biomedical Sciences, University of Strathclyde, Glasgow, United Kingdom

* benjamin.pickard@strath.ac.uk

## Abstract

Conventional expression studies quantify messenger RNA (mRNA) transcript levels gene-by-gene. We recently showed that protein expression is modulated at a global scale by amino acid availability, suggesting that mRNA expression levels might be equivalently affected. Through re-analysis of public transcriptomic datasets, it was confirmed that nucleobase supply interacts with the specific demands of mRNA $A+U:C+G$ sequence composition to shape a global profile of expression, which can be quantified as a gradient of average expression change by average composition change. In mammals, each separate organ and cell-type displays a distinct baseline profile of global expression. These profiles can shift dynamically across the circadian day and the menstrual cycle. They are also significantly distorted by viral infection, multiple complex genetic disorders (including Alzheimer's disease, schizophrenia, and autoimmune disorders), and after treatment with 115 of the 597 chemical entities analysed. These included known toxins and nucleobase analogues, but also many commonly prescribed medications such as antibiotics and proton pump inhibitors, thus revealing a new mechanism of drug action and side-effect. As well as key roles in disease susceptibility, mRNAs with extreme compositions are significantly over-represented in gene ontologies such as transcription and cell division, making these processes particularly sensitive to swings in global expression. This may permit efficient, *en bloc* transcriptional reprogramming of cell state through simple adjustment of nucleobase proportion and supply. It is also proposed that this mechanism helped mitigate the loss of essential amino acid synthesis in higher organisms.

In summary, global expression regulation is invisible to conventional transcriptomic analysis, but its measurement allows a useful distinction between active, promoter-mediated gene expression changes and passive, cell state-dependent transcriptional competence. Linking cell metabolism directly to gene expression offers an entirely new perspective on evolution, disease aetiopathology (including gene x environment - GxE - interactions), and the nature of the pharmacological response.

provided the original author and source are credited.

**Data availability statement:** All relevant data are within the manuscript and its Supporting Information files.

**Funding:** The author(s) received no specific funding for this work.

**Competing interests:** The author has declared that no competing interests exist.

## Introduction

To orchestrate complex programmes of embryonic development and adult organism homeostasis, static chromosomal DNA must generate highly dynamic, promoter-regulated profiles of messenger RNA (mRNA) and, subsequently, functional protein expression. In contrast to the documented complexities of promoter regulation, the actual molecular syntheses of mRNA and protein are largely ignored as a straightforward and frictionless transfer of encoded information. However, both processes are vast enzyme-mediated polymerisation reactions taking place within partially sealed vessels (the nucleus, the cell) in which substrates (here, nucleobases or amino acids) are finite. Substrate concentration is a key determinant of biochemical reaction rate and, therefore, warrants consideration as a potential regulator of expression.

We recently reported that situational shortages in amino acid supply from diet or biosynthesis modulate global protein expression levels, agnostic of mRNA abundance and, therefore, promoter activity [1]. In essence, this global protein expression can be thought of as the 'wood' to the 'trees' of individual protein expression, offering a high-level diagnostic of cell state. Here, the natural follow-on question is posed: does a reduced availability of RNA nucleobases (for example through disease or drug treatment) influence global gene expression levels independently of promoter activity? The key to understanding global expression regulation is that any deficit in substrate supply will not affect all mRNAs or proteins equally. For those with extreme sequence composition, synthesis will be disproportionately constrained, reducing relative expression level within the cell. To reveal global protein expression effects, we stratified proteomic expression data by the relative composition of the three nutritional subgroups of amino acids (essential amino acids, EAA, required from diet; non-essential amino acids, NEAA, synthesised within the body; and conditionally essential amino acids, CEAA, inefficiently synthesised) within each protein. This was a logical choice because the subgroups reflect natural dietary/biosynthetic constraints that could have driven evolutionary adaptation. For mRNA expression, an analogous constraint would be insufficient supply of purine (A/G) or pyrimidine (C/U) nucleobases through altered biochemical synthesis or salvage [2]. However, the prevailing view has always been that nucleobase supply is tightly and co-ordinately regulated to preserve availability for DNA and RNA synthesis [3]. Nevertheless, a re-examination of 45 publicly available transcriptomic datasets was carried out using a new approach that examines the global influence of mRNA nucleobase composition on expression level. This not only acts as a proxy for the direct quantification of free nucleobase availability, but also identifies those genes and biological processes most affected as global expression profiles shift. Using this approach, global expression profiles were compared across different tissues, disease states, and drug treatments.

## Results

### Detecting global changes in gene expression profile

Gene transcript levels were examined at a global level in multiple publicly available gene expression datasets. Fig 1 outlines the concept and process used here to

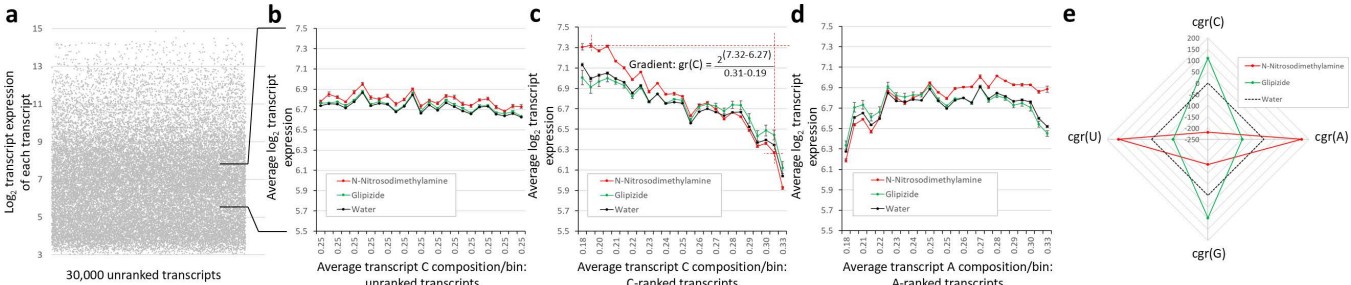

**Fig 1. Stratification of expression data by mRNA base composition reveals transcript level regulation at a global scale that can be quantified by the gradient of the profile.** *Rat liver gene expression changes (Gene Expression Omnibus, GEO dataset GSE57815 [4]) after in vivo treatment with N-nitrosodimethylamine or glipizide are used here to illustrate the quantification of global expression profile. The complexity of individual transcript expression (a) can be removed by binning transcripts and plotting each bin's average gene expression (b). A higher expression baseline after N-nitrosodimethylamine treatment may represent a failure to normalise the dataset before public deposition of data. Both intrinsic and drug-induced global effects on expression, likely the result of limiting nucleobase availability, become apparent when transcripts are assigned to bins according to increasing nucleobase composition; for example by nucleobase C (c) or A (d), and then average expression per bin plotted (with standard error of the mean, N = 3). In (c), the gradient of expression change (gr(C)) for N-nitrosodimethylamine is shown calculated between the 2nd to the 24th C bin, chosen to avoid extreme outlier effects. By subtracting the control gradient, a corrected gradient, cgr(base), can also be calculated for both drugs. A radar plot allows drug effects on cgr for all four bases to be viewed simultaneously (e).*

achieve this goal and to visualise outcomes. The typical high dynamic range ($2^3$ to $2^{15}$) of individual gene expression is illustrated for over 30,000 transcripts detected by microarray in the rat liver [4] when treated with water as control (Fig 1a). In a 'sham' analysis using the same data, the average transcript expression was calculated and plotted within 25 equal-sized bins each containing hundreds of randomly assigned transcripts (Fig 1b). This generated a simplified profile of expression within the control liver and, in addition, within livers treated with example agents, N-nitrosodimethylamine and glipizide. As expected, the plot is largely flat. However, when the same transcripts were ranked and binned along the x-axis according to increasing C nucleobase composition (**Methods,** Fig 1c), a substantial influence of composition on average bin expression level became apparent. A similar approach with A nucleobase composition also showed an influence on average expression level (Fig 1d).

There is an intrinsic mRNA nucleobase composition effect on expression visible in control samples: a move from lowest to highest C composition bin is accompanied by a halving of average transcript expression level. For A, it is more complex with a general positive influence on expression except at the extremes of composition. N-nitrosodimethylamine and glipizide treatments further distort the baseline global expression, but in opposing directions, resulting in anticlockwise and clockwise gradient changes, respectively, when ranking by A composition. To obtain numerical values for the action of a drug (or disease, see below) on expression, the gradient of expression change was calculated, as shown in Fig 1c for nucleobase C: **gr(C)**. The gradient for water, or appropriate solvent control, can be subtracted from this value to give a corrected value, **cgr(C)**, that offers some independence from microarray chip-specific effects or undocumented expression normalisation procedures. It is hypothesised that changing gradients indicates the impact of a drug on nucleobase availability for incorporation into growing transcripts. Here, N-nitrosodimethylamine appears to be reducing C availability, further increasing constraint on transcripts with higher proportions of C and, thus, decreasing their expression level. Those transcripts with reduced C composition show *relatively* enhanced expression when treated with N-nitrosodimethylamine. In Fig 1e, corrected gradient values for the two drugs and water control are shown on a radar plot, allowing drug influences on all four mRNA nucleobases to be visualised simultaneously, with the control gradient represented by the 0 contour (dashed line). This form of plot highlighted an unexpected finding - that drug-induced perturbations in gene expression profile do not vary in dimensions aligned with a simple purine (gr(A), gr(G)) or pyrimidine (gr(C), gr(U)) biochemical pathway disruption. Instead, global gene expression distortions primarily move along gr(C) correlated with gr(G),

and gr(A) correlated with gr(U) directions. Henceforth, gr(C) and gr(A) alone are plotted as representatives of these two dimensions, and any shift in transcriptional profile is termed as being in either the 'C-A+' or 'C+A-' direction.

## Nucleobase composition differences between mRNA subregions underpin global transcriptional control

The subregional features of mRNA composition were examined as a potential source of the A+U and C+G pairings of gradient movement. Within 110,962 human mRNA transcripts (protein-coding component of the GENCODE v44), the average nucleobase frequencies across the human transcriptome are very similar: 0.249 (C), 0.257 (A), 0.256 (G), and 0.238 (U) (Fig 2a) and average transcript length is 2,395 nucleotides. However, analysis of mRNA subregions revealed distinct base profiles: the 5' untranslated region (UTR) shows enrichment for C and G bases (Fig 2b), likely correlating with promoter CpG island proximity; the protein-coding sequence (coding DNA sequence, CDS), constrained by its coding function, shows no clear bias (Fig 2c); while the 3' UTR shows enrichment for A and U bases (Fig 2d). Shorter mRNAs are disproportionately (C+G) rich on average (Fig 2e), but longer ones, by virtue of their proportionally greater CDS and 3'UTR contribution to length (Fig 2f), tend to greater (A+U) richness. These pairings of mRNA nucleobase composition frequencies match the nucleobase pairings observed for the dimensions of global transcriptional change. This suggests a model whereby nucleobase availability (supply) may be variable in nature and direction, but transcript composition (demand) dictates its consequences on expression, restricting global profile changes to the C+A- or C-A+ type.

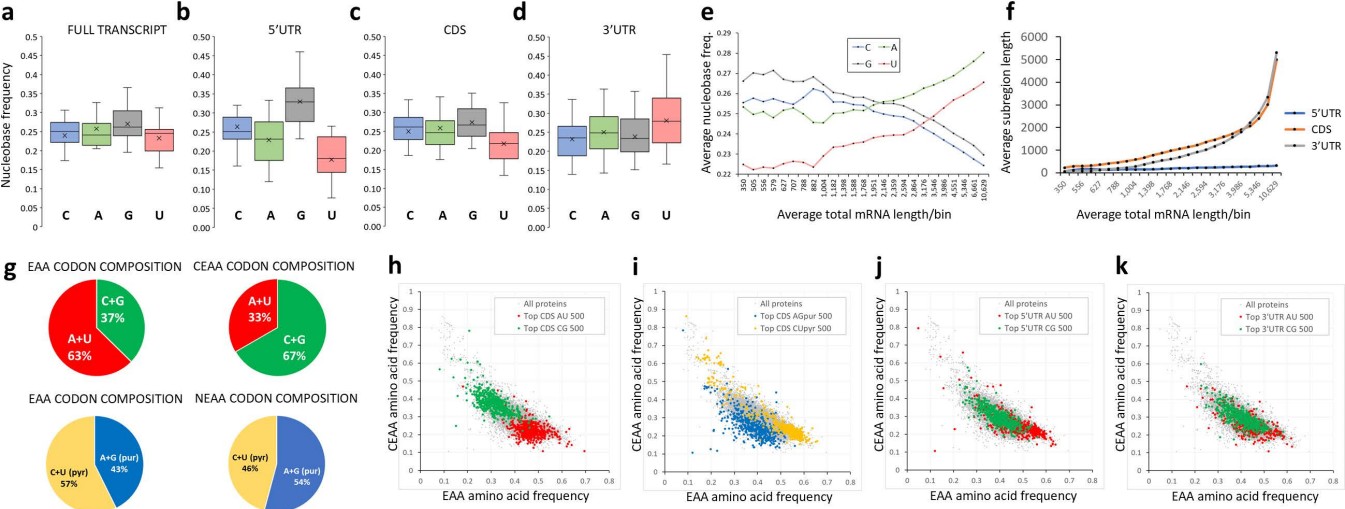

**Fig 2. Sequence composition analysis identifies distinct A +U and C+G base frequencies within subregions of mRNA transcripts – with effects on protein coding potential.** *The proportions of the four nucleobases were quantified in all human full-length mRNA transcripts (a) and then by subregion: 5'UTR sequences (b), CDS coding sequences (c), and 3'UTR sequences (d) with results displayed as box-and-whisker plots. The differences in nucleobase composition between subregions show correlations between C and G (both decreasing in frequency) and, separately, between A and U bases (increasing) as the length of transcript increases (e). This length effect in nucleobase representation is explained by total transcript length increases deriving primarily from increases in A+U enriched CDS and 3'UTR sequences (f). In (g), the sets of codons for the nutritional groupings of amino acids show biases in nucleobase frequency (EAA: Essential amino acids; CEAA: conditionally essential amino acids; NEAA: non-essential amino acids). The consequences of these biases are revealed in scatterplots of all proteins (grey), plotted according to EAA and CEAA composition (h-k). The top 500 mRNAs for enrichment of A+U or C+G nucleobases in the CDS code for proteins enriched in EAAs (red) or CEAAs (green), respectively (h). A similar but weaker effect is seen for extreme purine- and pyrimidine-rich mRNAs and NEAAs (blue) and EAAs (yellow) in their proteins, respectively (i). mRNA-protein composition correlations are not only driven by direct action of CDS sequences – extreme A+U composition biases in the 5'UTR sequences (j) and, to a lesser extent, 3'UTR sequences (k) also influence EAA/CEAA coding biases in proteins, indicating that detection of nucleobase supply (and impact on transcriptional competence) is transcript-wide.*

From first principles, CDS sequences with extreme nucleobase compositions would be expected to encode proteins with a biased amino acid profile. However, this biased profile unexpectedly aligned with the nutritional groupings of amino acids. This is a consequence of the A + U and C + G pairings having very different relative representations within the codons for the 9 essential amino acids (EAA) and the codons for the 6 conditionally essential (CEAA) amino acids (Fig 2g **top**). To a lesser extent this is also true for the comparison of C + U (pyrimidine) and A + G (purine) pairings, which showed representational differences between EAA codons and those for the 5 non-essential amino acids (NEAA) (Fig 2g **bottom**). The impact of this connection between nucleobase composition and nutritional amino acid groupings is that mRNAs of extreme composition (for example, the 500 most A + U-rich or C + G-rich in the CDS subregion) encode proteins that are highly EAA- or CEAA-enriched, respectively (p-value comparing EAA composition between the 500 top A + U nucleobase enriched proteins against all proteins: $1.1 \times 10^{-69}$, p-value comparing CEAA composition between the 500 top C + G enriched proteins against all proteins: $8.0 \times 10^{-100}$) (Fig 2h). Purine/pyrimidine-rich CDS sequences showed less significant coding discrimination between EAA and NEAA groupings (Fig 2i).

Equally unexpectedly, the A + U influence on amino acid composition was not limited to the direct protein coding effects of the CDS region. Compositionally extreme non-coding 5'UTR or 3'UTR sequences, considered in isolation, also showed a significant influence on associated protein amino acid compositions (5'UTR effect p-values: EAA $2.1 \times 10^{-64}$, CEAA $1.2 \times 10^{-27}$; 3'UTR effect p-values: EAA $2.8 \times 10^{-17}$, CEAA $1.7 \times 10^{-19}$) (Fig 2j,k). This influence was further confirmed by the modest positive correlations observed between sequence A + U composition frequencies within the isolated CDS sequences and the adjoining 5'UTR ($R^2$ 0.12) or 3'UTR ($R^2$ 0.15) sequences. This phenomenon did not occur with the purine/pyrimidine pairings. To summarise this section, nucleobase composition variation is largely limited to the A + U and C + G dimensions by a mRNA subregion interaction with length, and this has undergone evolutionary selection along the *entire* length of mRNA transcripts, presumably to increase sensitivity to nucleobase supply changes in a manner that affects both global mRNA expression as well as the nutritional/biosynthetic amino acid profile of encoded proteins.

## Global gene expression profiles are perturbed by many drugs/chemical entities

Thirteen publicly available drug-profiling gene expression datasets from human, mouse and rat (Gene Expression Omnibus, [5,6]) were chosen for analysis, comprising induced pluripotent cell lines (partially differentiated down neural progenitor or cardiomyogenic lineages), transformed cell lines, and multiple tissue samples from rodents treated *in vivo* (study details in S1 Table). In total, data from almost 1,000 individual treatments was reassessed, comprising 597 unique chemical entities including nucleobase analogues, commonly prescribed medications, bioactive molecules, and potential environmental toxins. The cgr(base) quantification approach was applied to all 13 studies and each drug's actions visualised by plotting cgr(C) against cgr(A). As an example, Fig 3a plots global gene expression profile changes in rat bone marrow after treatment with 76 drugs/agents (GSE59894, [4]). The majority of drugs are positioned on a top-left to bottom-right diagonal, bidirectionally extending from the origin. However, a minority show an 'asymmetric' effect, placing them above or below the diagonal, perhaps indicating a more complex nucleobase deficit. Radar plots allow full visualisation of nucleobase influences from example drugs with symmetric (Figs 3b **C-A+ and 3c C + A-**) and asymmetric (Fig 3d) effects on global expression.

The published toxicities and historical market withdrawals identified for many extreme outlier drugs indicated that the magnitude of cgr(X) in any direction is probably limited by biological viability and adverse effects. Thus, to be categorised as an outlier, an agent was only required to exceed one standard deviation of the solvent control mean along at least two of the four nucleobase dimensions (approximately indicated by the grey oval in Fig 3).

Bleomycin A2 (DNA cleavage in cancer treatment), urethane (anaesthetic/carcinogen), and mitomycin C (DNA cross-linking/alkylating action in cancer treatment) were chosen from the GSE59894 study (Fig 3) for further study, having very diverse molecular structures, unrelated and non-specific pharmacological actions, and extreme outlier placements on the cgr(base) plot (S1a Fig). Firstly, to test the nucleobase supply hypothesis without direct quantification of free

nucleobases in the tissue samples, it was reasoned that the relative frequencies of nucleobases sequestered within the transcribed cellular mRNA population would be at equilibrium with the free pool of nucleobases and could, therefore, be used to estimate changes in availability after drug treatment. The relative frequency of each sequestered nucleobase was obtained by summing the expression levels of all transcripts after each had been multiplied by its nucleobase frequency. S1b Fig shows that the C nucleobase component of the mRNA population was reduced after bleomycin A2 or urethane treatment when compared to water control or to mitomycin C treatment. The opposite effect was seen for A nucleobases. While these changes are small in scale, they are statistically significant, supporting a supply effect. The second analysis sought to determine if global effects on expression are universal in consequence, governed only by direction of profile change and transcript composition. Specifically, overlaps in upregulated gene expression were examined across two pharmacologically distinct drug actions (bleomycin A2 and urethane) united only by C-A+ profile, and one drug (mitomycin C) with opposing C+A- profile (Fig 3). Of the 100 transcripts most highly upregulated after bleomycin A2 treatment, 75

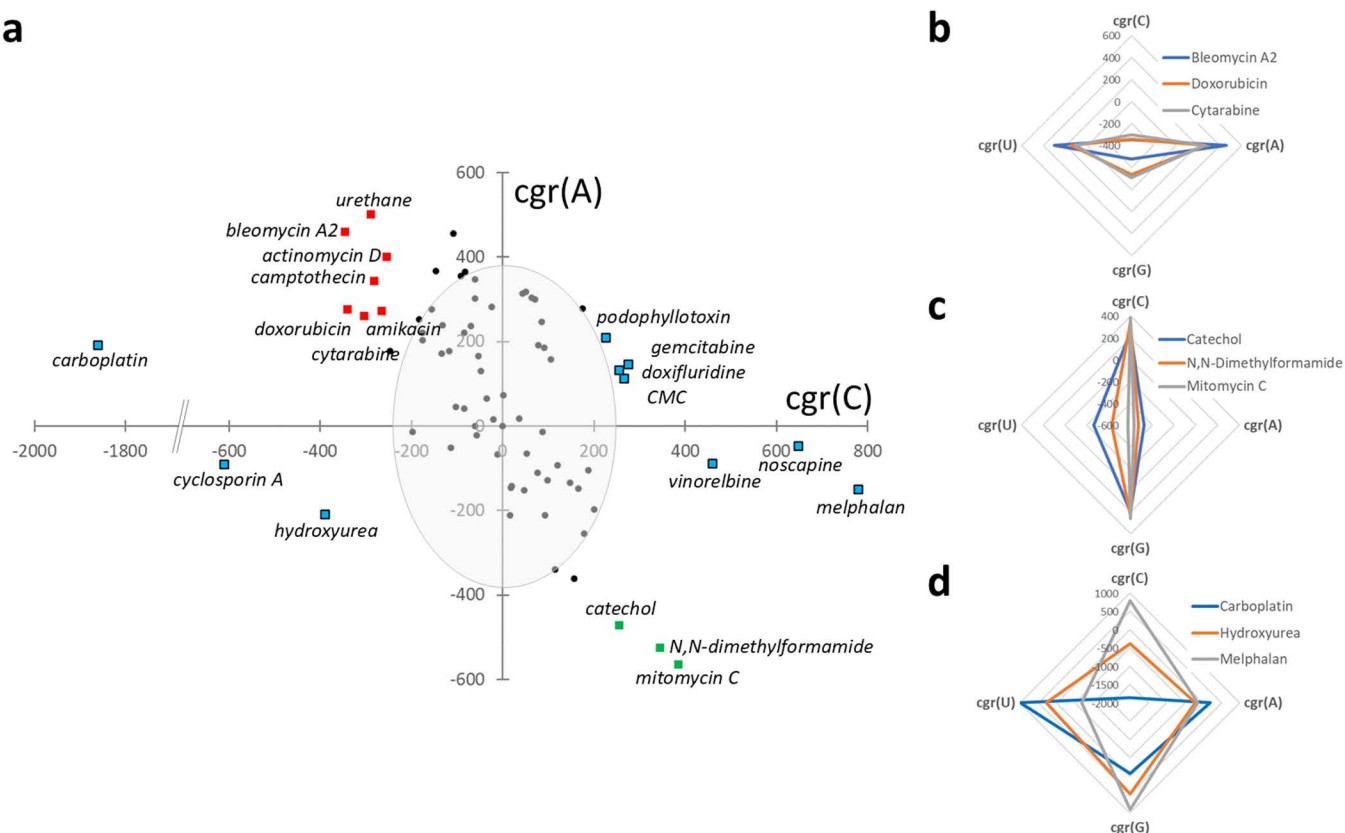

**Fig 3. A gradient plot of a representative study from the full drug screen shows outlier molecules with substantial in vivo effects on the global transcriptional profile of rat bone marrow cells.** Global gene expression profile changes in rat bone marrow after treatment with 76 drugs/agents (GSE59894, [4]). Data are plotted on a cgr(C) vs cgr(A) scatterplot where cgr represents the control corrected gradient of the expression profile for that base (a). The distribution is centred around the origin but extends primarily diagonally, top-left to bottom-right. Data points represent the average of the biological replicates of expression quantification for each drug, with replicates typically numbering twelve. One standard deviation of the control samples is indicated by the grey oval. Green squares indicate outlier (C+A-) drugs inducing substantial expression increases in transcripts with higher proportion of C bases and concomitant decreases in those with increased A proportions. Red squares highlight drugs with outlier (C-A+) effects in the opposite direction. Although the majority of drug treatments caused symmetric profile changes along the diagonal C-A+ to C+A- axis, several drugs deviated from symmetry, as denoted by blue squares. Carboplatin has the most extreme, asymmetric C- shift requiring the X-axis to be cropped. Illustrative radar plots of symmetric effect drugs (Figs 3b **C-A+** and 3c **C+A-**), and asymmetric effect drugs (Fig 3d) cause characteristic global expression profile perturbations.

were identical to the 100 most highly upregulated urethane treatment transcripts, but only 4 or 2 were shared with those most highly upregulated by mitomycin C, with 11 common to all three treatments. The expression levels of the 100 most highly upregulated bleomycin A2 transcripts were tightly correlated with their expression levels after urethane treatment ($R^2 = 0.96$), but less so with their expression after mitomycin C treatment ($R^2 = 0.51$) (S1c Fig). These 100 transcripts have an average C composition frequency of 0.243 (significantly lower than the 100 most downregulated transcripts, average 0.266, t-test p-value $1.14 \times 10^{-05}$) and an A composition frequency of 0.261 (significantly higher than the 100 most downregulated transcripts, average 0.239, t-test p-value $2.34 \times 10^{-05}$). Together, these results support a model of transcript expression with three components: a baseline component of transcript expression that is tissue-specific and promoter-driven (hence, the positive expression correlations throughout), a substantial, directional, and universal (promoter independent) nucleobase supply-driven component after outlier drug treatment, and a modest drug-specific, promoter-driven expression component.

Fig 4 lists the identities and the clinical and molecular properties of the 131 treatments (115 unique entities, 19.3% of total) across the 13 studies that caused outlier global expression, grouped by radar plot shape (symmetric C-A+ or C+A- directions, or asymmetric). Numerical data from all 13 studies are available in tabulated form in S2 Table.

Importantly, nucleobase analogues and antimetabolite drugs are present within Fig 4, further indicating that a nucleo-base supply mechanism is consistent with the global perturbations observed. Independent replications add to the robustness of the findings. The analogue cytarabine and alkylating agent mitomycin C were both outliers in three independent studies; and cisplatin, N-nitrosodimethylamine, catechol, CMC (carboxymethyl cellulose, a thickening agent used in food, cosmetics, and pharmaceuticals), methapyrilene, hydroxyurea, trichostatin A, imatinib, cyclophosphamide, lipopolysac-charide, melphalan, and methotrexate, were outliers in two independent studies. However, the direction of transcriptional distortion for specific drugs was occasionally different between tissues, such as in the case of methotrexate showing a C-A+ shift in the rat spleen but C+A- shift in human iPS cells (cardiomyogenic differentiation). Moreover, cisplatin has profound C+A- axis effects on expression in rat hepatocytes and human HepG2 cells, but little effect in rat kidney, bone marrow, or spleen. Tissue-/experiment-selective effects were observed for many other drugs, perhaps reflecting differing *in vivo* absorption, distribution, metabolism, and excretion properties between tissues and cells.

Some chemical structure/function correlations were clear within the data. Analogue doxifluridine and its metabolite, fluorouracil, were independently identified as outliers, as were the epigenetic modifiers trichostatin A and chemical deriv-ative, vorinostat. Gefitinib and imatinib, rather distant cousins within the EGFR inhibitor family, were both outliers but erlo-tinib (structurally very similar to gefitinib) was not. Proton pump inhibitors, lansoprazole and pantoprazole, were outliers. Related rabeprazole was similar in its direction of transcriptional effect, if not the magnitude required to be defined as an outlier, but a fourth member of the family, omeprazole, exhibited almost no effect on global transcription.

## Tissues show distinct baseline global transcriptional profiles

In the analysis above, tissues and cell types differed from each other in both the proportion of drug treatments that per-turbed global expression profiles as well as the magnitude of those perturbations. The spleen showed the most robust outlier population, while kidney showed the least. This prompted an analysis of baseline tissue global expression profiles as a possible explanation. Three independent transcriptomic datasets of mouse organ/tissue/cell line expression profiles (GSE24207, GSE10246, GSE9954) were assessed (S1 Table ). All three produced largely similar gr(base) profiles, with GSE9954 shown in Fig 5. Among these, mouse spleen showed an extreme C-A+ distortion, perhaps rendering it more susceptible to transcriptional drug effects, whereas the kidney is more centrally located, perhaps making it more resistant to change. A key observation is that fetal tissues and ovary, together with embryonic stem, transformed, and haemato-poetic cell lines occupy an extreme C-A+ region of the plot, suggesting that differentiation state or proliferative potential correlate with profile position on the bidirectional axis. The radar plot inset confirms the reduction in gr(C) in undifferenti-ated tissues, but also highlights a large increase in gr(G). Testis, despite its extraordinarily offset outlier position (due to its

**Fig 4.** Chemical drug/entity treatments grouped by global transcription expression profile effect.

| GLOBAL EFFECT DIRECTION | SPECIES/TISSUE | AGENT | OTHER | MECHANISM |
|---|---|---|---|---|
| Asymm. | Rat. Hepatocytes | Lomustine | | Alkylating agent |
| Asymm. | Rat. Bone Marrow | Melphalan | | Alkylating agent |
| Asymm. | Rat. Hepatocytes | Carmustine | | Alkylating agent |
| Asymm. | Rat. Bone Marrow | Gemcitabine | | Analogue dCTP |
| Asymm. | Rat. Bone Marrow | Hydroxyurea | Sickle cell drug | Antimetabolite |
| Asymm. | Rat. Spleen | Hydroxyurea | Sickle cell drug | Antimetabolite |
| Asymm. | Rat. Liver | Fluorouracil | | dTMP synthesis inhibitor |
| Asymm. | Rat. Liver | Imatinib | | EGFR inhibitor |
| Asymm. | Rat. Bone Marrow | Carboplatin | | Metal |
| Asymm. | Rat. Bone Marrow | Vinorelbine | | Microtubule inhibitor |
| Asymm. | Rat. Bone Marrow | Doxifluridine | | Prodrug for Fluorouracil |
| Asymm. | Rat. Bone Marrow | Noscapine | Anti-tussive | Sigma opioid receptor/Microtubule inhibitor |
| Asymm. | Hom. iPS (neural progenitor cells) | Trichostatin A | Anti-fungal | HDAC inhibitor |
| Asymm. | Hom. iPS (neural progenitor cells) | Vorinostat | | HDAC inhibitor |
| Asymm. | Rat. Bone Marrow | Podophyllotoxin | | Plant extract |
| Asymm. | Rat. Spleen | Famciclovir | | Analogue G |
| Asymm. | Rat. Liver | Roflumilast | | PDE4 inhibitor |
| Asymm. | Rat. Spleen | Meropenem | | Beta lactam |
| Asymm. | Rat. Bone Marrow | Cyclosporin A | Immunosuppressant | Calcineurin inhibitor? |
| Asymm. | Rat. Brain | Caffeine | Stimulant | Adenosine receptor inhib |
| Asymm. | Rat. Intestine | Bisacodyl | Laxative | Fluid and salt secretion |
| Asymm. | CMC | CMC | | Formulation component |
| Asymm. | Rat. Hepatocytes | Lansoprazole | GERD treatment | Proton pump inhibitor |
| Asymm. | Rat. Hepatocytes | Pantoprazole | GERD treatment | Proton pump inhibitor |
| Asymm. | Rat. Hepatocytes | Monocrotaline | Toxin | Plant alkaloid |
| C-A+ | Rat. Spleen | Altretamine | | Alkylating agent |
| C-A+ | Rat. Spleen | Melphalan | | Alkylating agent |
| C-A+ | Rat. Liver | Mitomycin C | | Alkylating agent |
| C-A+ | Rat. Liver | Procarbazine | | Alkylating agent |
| C-A+ | Rat. Spleen | Methotrexate | | Antimetabolite: folate |
| C-A+ | Rat. Heart | Sulindac | | Arylalkanoic acid |
| C-A+ | Rat. Bone Marrow | Bleomycin A2 | | DNA damage |
| C-A+ | Rat. Spleen | Imatinib | | EGFR inhibitor |
| C-A+ | Hom. iPS (cardiomyogenic) | Trichostatin A | Anti-fungal | HDAC inhibitor |
| C-A+ | Hom. iPS (cardiomyogenic) | Vismodegib | | Hedgehog signalling inhibitor |
| C-A+ | Rat. Bone Marrow | Actinomycin D | | Microtubule inhibitor |
| C-A+ | Rat. Liver | Vinblastine | | Microtubule inhibitor |
| C-A+ | Rat. Bone Marrow | Camptothecin | | Topoisomerase inhibitor |
| C-A+ | Rat. Bone Marrow | Doxorubicin | | Topoisomerase inhibitor |
| C-A+ | Rat. Spleen | Cytarabine | | Analogue C/U |
| C-A+ | Rat. Spleen | Cytarabine | | Analogue C/U |
| C-A+ | Rat. Liver | Cytarabine | | Analogue C/U |
| C-A+ | Rat. Heart | Cyclophosphamide | | Alkylating agent |
| C-A+ | Hom. HepG2 cells | Adefovir | | Analogue A |
| C-A+ | Hom. iPS (cardiomyogenic) | Favipiravir | | Analogue A/G? |
| C-A+ | Rat. Kidney | Stavudine | | Analogue T |
| C-A+ | Rat. Spleen | Penciclovir | | Analogue G |
| C-A+ | Rat. Spleen | Tranilast | Allergy | Unknown target |
| C-A+ | Hom. iPS (neural progenitor cells) | Emetine | Vomiting induction | Unknown target |
| C-A+ | Rat. Liver | Zileuton | Asthma treatment | 5-lipoxygenase inhibitor |
| C-A+ | Rat. Bone Marrow | Amikacin | | Aminoglycoside |
| C-A+ | Rat. Kidney | Streptomycin | | Aminoglycoside |
| C-A+ | Rat. Kidney | Allopurinol | Gout | Analogue PUR |
| C-A+ | Rat. Liver | Aminosalicylic Acid | | Antimetabolite: folate |
| C-A+ | Rat. Liver | N-Nitrosodimethylamine | | Arg synthesis, DNA replic inhib |
| C-A+ | Mus Liver (male) | N-Nitrosodimethylamine | | Arg synthesis, DNA replic inhib |
| C-A+ | Rat. Spleen | Tazobactam | | Beta lactam |
| C-A+ | Rat. Spleen | Aztreonam | | Beta lactam |
| C-A+ | Rat. Liver | Methapyrilene | | Carcinogen |
| C-A+ | Rat. Hepatocytes | Methapyrilene | | Carcinogen |
| C-A+ | Rat. Liver | N-Nitrosodiethylamine | | Carcinogen |
| C-A+ | Rat. Bone Marrow | Urethane | Anaesthetic | Carcinogen |
| C-A+ | Rat. Liver | Salicylic Acid | | COX inhibitor |
| C-A+ | Rat. Kidney | Lead (II) Acetate | | Metal |
| C-A+ | Rat. Liver | Albendazole | | Microtubule inhibitor |
| C-A+ | Rat. Muscle | Guanethidine | | Norepinephrine release inhibitor |
| C-A+ | Hom. iPS (neural progenitor cells) | Anisomycin | | Peptidyltransferase inhibitor |
| C-A+ | Rat. Kidney | Bacitracin | | Polypeptide |
| C-A+ | Mus Liver (male) | Wy-14643 | | PPAR alpha activator |
| C-A+ | Rat. Heart | Bezafibrate | | PPAR alpha activator |
| C-A+ | Rat. Liver | Nafenopin | | PPAR alpha activator, liver carcinogen |
| C-A+ | Hom. HepG2 cells | Tetracycline | | Protein synth. inhibitor |
| C-A+ | Hom. HepG2 cells | Ethinyl estradiol | Estrogen | Female combined contraceptive pill |
| C-A+ | Rat. Spleen | Isoniazid | | Tuberculosis treatment |
| C-A+ | Hom. HepG2 cells | Nifedipine | Dihydropyridine | Calcium channel blocker |
| C-A+ | Rat. Hepatocytes | Risperidone | | Ser./Dop. Rec. antagonist |
| C-A+ | Rat. Hepatocytes | Venlafaxine | | Ser./Nor. reuptake inhibitor |
| C-A+ | Rat. Intestine | Cisapride | GERD treatment | Serotonin Rec agonist |
| C-A+ | Rat. Hepatocytes | Sparteine | Anti-arrhythmic | Sodium channel block |
| C-A+ | Hom. iPS (cardiomyogenic) | Carbamazepine | | Sodium channel block |
| C-A+ | Rat. Hepatocytes | Diphenhydramine | | Sedative effects |
| C-A+ | Rat. Heart | Loratadine | | Sedative effects |
| C-A+ | Rat. Brain | Piracetam | Cognition/Sickle cell | GABA derivative |
| C-A+ | Rat. Brain | Nicotine | Stimulant/anxiolytic | Acetylcholine Rec agonist |
| C-A+ | Mus Liver (male) | Thioacetamide | | Carcinogen |
| C-A+ | Rat. Spleen | Bisphenol A | | Xenoestrogen |
| C-A+ | Rat. Spleen | Catechol | | Possible carcinogen |
| C-A+ | Rat. Liver | Allyl Alcohol | | Metabolised to toxic acrolein |
| C-A+ | Rat. Liver | Carbon Tetrachloride | | Possible carcinogen and hepatotoxin |
| C-A+ | Rat. Kidney | 2-Amino-4-Nitrophenol | | Kidney toxicity |
| C-A+ | Rat. Liver | Lipopolysaccharide | Endotoxin | Bacterial |
| C-A+ | Rat. Liver | Lipopolysaccharide | Endotoxin | Bacterial |
| C-A+ | Hom. iPS (neural progenitor cells) | Lycorine | Toxin | Plant alkaloid |
| C-A+ | Hom. iPS (cardiomyogenic) | Retinol | Vitamin A | Fat soluble |
| C-A+ | Rat. Kidney | Calcitriol | Vitamin D | Fat soluble |
| C+A- | Rat. Hepatocytes | Ifosfamide | | Alkylating agent |
| C+A- | Hom. HepG2 cells | Cyclophosphamide | | Alkylating agent |
| C+A- | Hom. HepG2 cells | Mitomycin C | | Alkylating agent |
| C+A- | Rat. Bone Marrow | Mitomycin C | | Alkylating agent |
| C+A- | Hom. iPS (cardiomyogenic) | Methotrexate | | Antimetabolite: folate |
| C+A- | Rat. Liver | Gefitinib | | EGFR inhibitor |
| C+A- | Rat. Heart | MLN-518 | | FLT3 RTK inhibitor |
| C+A- | Rat. Hepatocytes | Cisplatin | | Metal, purine adduct formation |
| C+A- | Hom. HepG2 cells | Cisplatin | | Metal, purine adduct formation |
| C+A- | Rat. Spleen | Etoposide | | Topoisomerase inhibitor |
| C+A- | Rat. Muscle | Azaribine | Psoriasis treatment | Metabolised to 6-AZA: Analogue PYR |
| C+A- | Rat. Spleen | Acyclovir | | Analogue G |
| C+A- | Hom. iPS (neural progenitor cells) | Perceptin | | AKA Cipralisant/GT-2331 |
| C+A- | Hom. HepG2 cells | Phenobarbital | | Barbiturate |
| C+A- | Rat. Heart | Atenolol | | Beta blocker |
| C+A- | Hom. iPS (cardiomyogenic) | Ampicillin | | Beta lactam |
| C+A- | Hom. iPS (cardiomyogenic) | Methicillin | | Beta lactam |
| C+A- | Rat. Kidney | Hydrocortisone | | Corticosteroid |
| C+A- | Rat. Brain | 5(-)-Carbidopa | | Dopa decarboxylase inhib |
| C+A- | Rat. Spleen | CMC | | Formulation component |
| C+A- | Rat. Heart | CMC/SDS/Povidone | | Formulation component |
| C+A- | Hom. HepG2 cells | Azathioprine | | Metabolised to 6-MP: Analogue PUR |
| C+A- | Hom. iPS (cardiomyogenic) | Magnesium Chloride | | Metal |
| C+A- | Rat. Spleen | Lead (IV) Acetate | | Metal |
| C+A- | Rat. Intestine | Loperamide | Anti-diarrhoea | Opioid |
| C+A- | Rat. Heart | Trimethadione | | Oxazolidinedione |
| C+A- | Rat. Heart | Pentoxifylline | Analgesic | Possible PDE inhibition |
| C+A- | Rat. Brain | Sibutramine | Appetite suppressant | Ser./Nor. reuptake inhibitor |
| C+A- | Rat. Intestine | Ondansetron | Anti-emetic | Serotonin Rec antagonist |
| C+A- | Rat. Brain | Oxcarbazepine | | Sodium channel block |
| C+A- | Hom. iPS (cardiomyogenic) | Paroxetine | | SSRI |
| C+A- | Rat. Liver | Glipizide | | Sulphonylurea |
| C+A- | Hom. iPS (cardiomyogenic) | Suloctidil | Vasodilator | Sulphur-containing aminoalcohol |
| C+A- | Rat. Liver | Propylthiouracil | Thyroid treatment | Thyroperoxidase inhibitor |
| C+A- | Rat. Hepatocytes | Sporidesmin A | Toxin | Fungal origin, facial eczema in ruminants |
| C+A- | Hom. iPS (cardiomyogenic) | Doxylamine | Allergy, Anti-emetic | Weak sedative effect |
| C+A- | Rat. Spleen | Catechol | | Possible carcinogen |
| C+A- | Rat. Bone Marrow | N,N-Dimethylformamide | | Possible carcinogen, CNS/organ toxicity |
| C+A- | Rat. Liver | Trichloroacetic Acid | Chemical peel | Herbicide/disinfectant |

(Category columns across the top: CHEMOTHERAPY, αVIRAL, αINFLAMMATORY, αHISTAMINE, αBIOTIC, αCONVULSANT, αPROTOZOAL, αHELMINTHIC, αDEPRESSANT, αPSYCHOTIC, αLIPEMIC, αDIABETIC, αHYPERTENSION, INDUSTRIAL, OTHER, MECHANISM, BMSuppr., TOXICITY, PREGN.)

**Fig 4. Chemical drug/entity treatments that induced a substantial change in global transcription expression profile.** *These 'outlier' (Methods) entities are grouped according to direction of effect on profile, experimental tissue, details of therapeutic mode/mechanistic action/chemical nature, and reported side-effects. (Abbreviations: BMSuppr., bone marrow suppression; PREGN., known cause of teratogenicity or pregnancy risk; Ser., serotonin; Dop., dopamine; Nor., norepinephrine; PDE, phosphodiesterase; SSRI, selective serotonin reuptake inhibitor).*

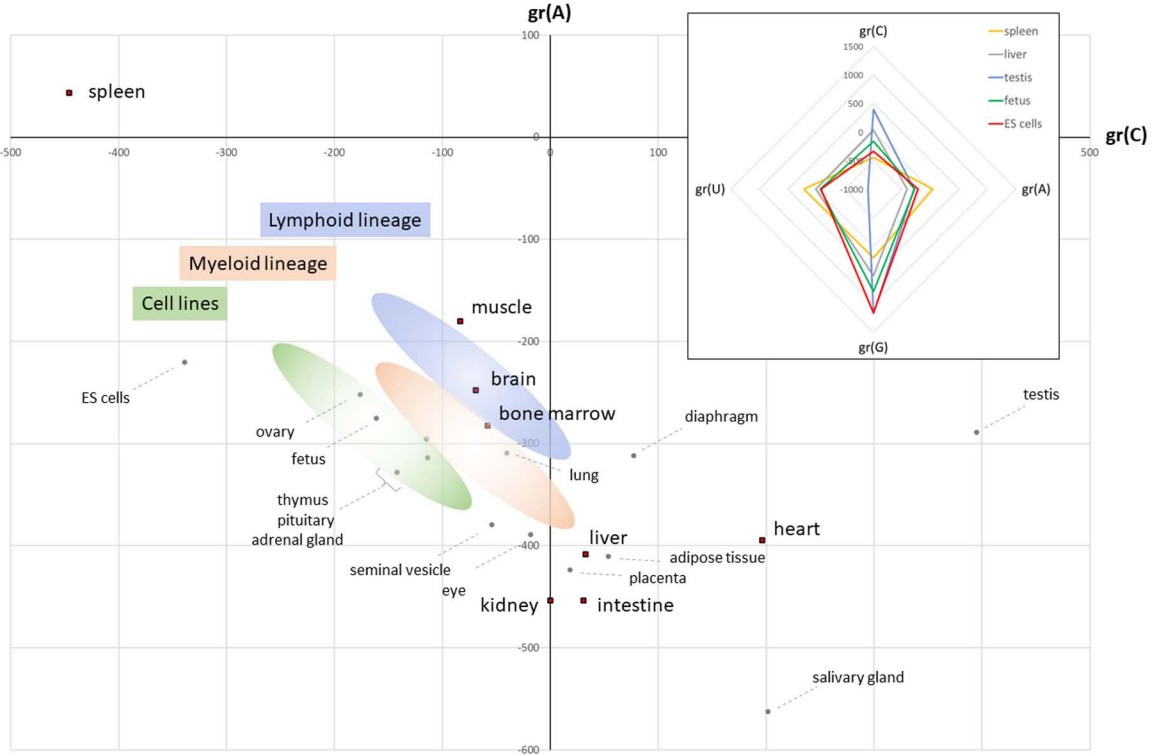

**Fig 5. A gradient plot indicates that mouse organs and cell types possess distinct global transcriptional profiles.** *Tissue global expression profile from study GSE9954. Tissues employed in the drug screen are indicated by red squares, highlighting the link between the spleen's extreme location and greatest perturbation by drugs. Coloured ovals indicate approximate positions for transformed cell lines and cells from the lymphoid and myeloid lineages, derived from GEO10246 data. The inset shows profile gradients for all four bases for four tissues and ES cells. The asymmetric location of testis in the main image is explained by the extreme negative gr(U) value in the radar plot.*

exceptionally negative gr(U)) also shares the large increase in gr(G). These data suggest that a tissue's baseline nucleobase availability might help it define its unique gene expression repertoire, drug sensitivity, and differentiation status.

## Physiological and pathological changes to global expression profiles

To determine if disease state or physiological change were associated with global expression profile shifts, gr(C) and gr(A) values were calculated from 30 publicly available array-based gene expression studies (S1 Table, Fig 6).

Lesch-Nyhan (LN) syndrome, a purine salvage disorder caused by *HPRT* gene mutation, presents with complex neurological and neuropsychiatric symptoms [7]. *In vitro* cellular knockdown data (GSE24345, Fig 6a) [8], when reanalysed, demonstrated a highly significant C+A- global transcriptome change – a genetic counterpart to the earlier nucleobase analogue drug treatments.

In blood samples, a significant profile shift in the same, C+A-, direction as LN syndrome was observed in a pair of studies (GSE63060/63061, Fig 6b,c) of blood samples from those diagnosed with Alzheimer's disease (AD) or mild cognitive impairment (MCI). The greater impact of MCI of profile may reflect early-stage pathology that becomes less detectable in full AD as neurons disappear. A similar shift was observed for one study (GSE111175, Fig 6d) of autism spectrum disorder (ASD), but two other ASD studies (GSE25507/42133, Fig 6e,f) failed to reach significance, even if their profiles trended in the same direction. Purine metabolism defects have been previously described for both AD [9] and ASD [10]. Together

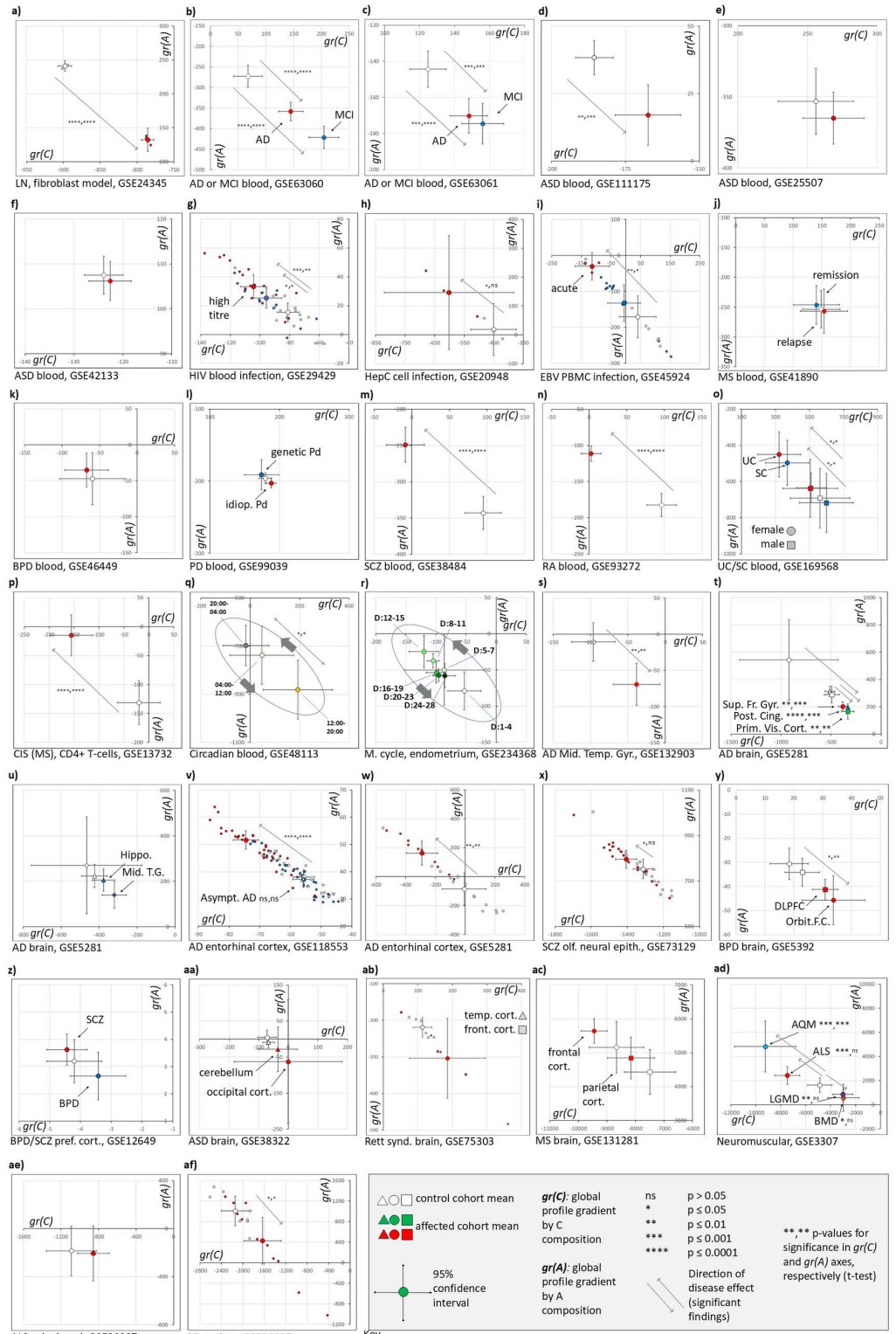

**Fig 6. Gradient plots indicate that viral infection and multiple complex genetic disorders significantly and bidirectionally perturb global transcriptomic profiles in blood, cells, and tissues.** *The panel of images plot gr(C) against gr(A) to represent the global transcriptional profile. Open shapes with error bars (95% confidence interval, conservative T-test statistic) represent mean values for uninfected or healthy control cohorts. Coloured*

*shapes indicate the corresponding means for the comparative infected/disease state. Statistically significant differences (two-tailed t-test) between healthy and disease gr(C) or gr(A) values, respectively, are indicated by asterisks, with arrows indicating effect direction. For viral response and some later tissue studies, when numbers of samples do not obscure interpretation, individual datapoints are plotted to illustrate variance. The GEO public expression data accession (GSExxx) and brief description are found under each plot, but full descriptions of sample sizes and analysis are in Supporting Information Table 1. For 5q and 5r, oval shapes indicate the cyclical passage of time (24-hr clock or D: days) for circadian and menstrual rhythms. For Rett syndrome analysis (ab), the circles represent the averages and CIs of 6 pooled affected or 6 pooled healthy control samples. (Abbreviations - AD: Alzheimer's disease (Asympt.: asymptomatic - pathology but no cognitive effects), LN: Lesch-Nyhan syndrome, MCI: mild cognitive impairment (prodromal AD), ASD: autism spectrum disorder, HIV: human immunodeficiency virus, HepC: hepatitis C virus, EBV: Epstein-Barr virus, PBMC: peripheral blood mononuclear cells, MS: multiple sclerosis, BPD: bipolar disorder, PD: Parkinson's disease (idiopathic or genetic), SCZ: schizophrenia, RA: rheumatoid arthritis, UC: ulcerative colitis, SC: a symptomatic form of inflammatory bowel disease that does not meet criteria for a diagnosis (for example, of UC), CIS: chronic isolation syndrome (prodromal MS), M. cycle: menstrual cycle, Mid. Temp. Gyr.: middle temporal gyrus of the brain, Sup. Fr. Gyr.: superior frontal gyrus of the brain, Post. Cing.: posterior cingulate cortex of the brain, Prim. Vis. Cort.: primary visual cortex of the brain, Hippo.: hippocampus of the brain, DLPC: dorsolateral prefrontal cortex of the brain, Orbit. **F. C.**: orbitofrontal cortex of the brain, Pref. cort.: prefrontal cortex of the brain, Front. cort.: frontal cortex of the brain, Temp. cort.: temporal cortex of the brain, AQM: acute quadriplegic myopathy, ALS: amyotrophic lateral sclerosis (motor neuron disease), LGMD: limb girdle muscular dystrophy, BMD: Becker muscular dystrophy, S. nigra: substantia nigra of the brain).*

with observations of a shared transcriptional profile between AD and LN [11], these findings suggest commonalities in biochemical disturbance are reflected at the global transcriptional level.

Infection by human immunodeficiency virus (HIV, GSE29429, Fig 6g), Hepatitis C virus (HepC, GSE20948, Fig 6h), and Epstein-Barr virus (EBV, GSE45924, Fig 6i) produced significant shifts in the opposite, C-A+, direction. Dose-dependent effects were observed in samples with higher viral titre (>100,000) for HIV, and in the acute primary infection state for EBV. This suggests that early-stage, active virion production might drain the host cellular C/G nucleobase supply available for transcription, a process that that may align with other more recognised 'host shutoff' mechanisms [12].

No significant global transcriptional changes were seen in blood samples for multiple sclerosis (MS, GSE41890, Fig 6j), bipolar disorder (BPD, GSE46449, Fig 6k), or Parkinson's disease (PD, GSE99039, Fig 6l). However, a significant C-A+ change for SCZ was observed in study GSE38485 (Fig 6m). Similarly, rheumatoid arthritis (RA, GSE93272, Fig 6n) presented with a significant C-A+ profile shift. Analysis of blood transcriptomics data from ulcerative colitis (UC) and symptomatic individuals not meeting UC diagnostic criteria (SC) (GSE169568, Fig 6o), showed trends in the C-A+ direction that were only statistically significant when females diagnosed with UC were compared to healthy control females, or when SC females were compared to SC males, indicating a potential influence of biological sex on phenotype. Crohn's disease did not show effects on global transcription. Clinically isolated syndrome (CIS), a precursor to full MS, produced a significant C-A+ global profile change in the CD4+T cell component of blood (GSE13732, Fig 6p). Finally, blood samples taken across the circadian day (GSE48113, Fig 6q) showed a C+A- shift during the afternoon/evening (12:00–20:00) and a C-A+ shift at night (20:00–0400). Mouse liver exhibits circadian fluxes in nucleobase biochemistry, especially purine metabolism, providing a necessary mechanistic link between circadian rhythm and the global expression changes observed [13]. Such shifts are likely to be a confounder for all the above transcriptomic studies in which blood drawing was not at a consistent time of day. They may also provide an explanation for aspects of chronotherapy – the time of administration of drugs listed within Fig 4 may synergise or antagonise with the circadian ebb and flow of changes in global expression profile [14–16].

Re-analysis of gene expression data from endometrial cell biopsies obtained across the menstrual cycle (GSE234368, Fig 6r) revealed another instance of cycling of the global gene expression profile: with the early follicular/proliferative phase (labels D1 through D15 indicating days post-menstruation) associated with a progressive C-A+ shift followed, after ovulation, by a C+A- return during the luteal/secretory phase (D16-D28). These changes may reflect the proliferation state of endometrial cells or the changing hormonal milieu (estrogen/FSH/LH/progesterone). The transcript encoding luteinizing hormone B chain (*LHB*) is in the top 0.5% of all transcripts for C-richness suggesting that its production in the pituitary gland may also be subject to global transcriptional control. Ethinyl estradiol is a component of the combined contraceptive pill and has a C-A+ effect in Fig 4. Similarly, bisphenol A, a toxic xenoestrogen identified with C-A+ effect

in the drug screen, is known to affect female reproductive health in numerous ways including menstrual irregularity and endometriosis [17].

In the assessment of other non-blood tissues in disease states, studies GSE36980 and GSE118553 provided no evidence for global expression changes associated with AD within the frontal cortex, temporal cortex, or hippocampus (S2 Fig). However, GSE132903 (Fig 6s) showed a significant C+A- shift for the middle temporal gyrus. GSE5281 (Fig 6t) showed significant C+A- changes within the visual cortex, the superior frontal gyrus, and the posterior cingulate cortex, with suggestive trends for the middle temporal gyrus and the hippocampus. Intriguingly, the AD global shift was statistically significant but reversed in direction (C-A+) within entorhinal cortex samples from two independent studies, GSE118553 (Fig 6v) and GSE5281 (Fig 6w). This brain region has unique characteristics that may correlate with it being one of the earliest locations for detectable AD pathology [18].

For SCZ, significant C-A+ changes were observed in biopsied olfactory neural epithelium (GSE73129, Fig 6x). For BPD, significant perturbations in the opposite direction to SCZ were observed for prefrontal cortex and orbitofrontal cortex (GSE5392, Fig 6y). However, no transcriptional changes were observed for either SCZ or BPD in another study of the prefrontal cortex (GSE12649, Fig 6z), although directionality of effect was maintained for both conditions.

For ASD, global profiles showed non-significant C+A- trends in the cerebellum and occipital cortex of a small study (GSE38322, Fig 6**aa**), and for the temporal cortex and frontal cortex in a study of the related condition, Rett syndrome (GSE75303, Fig 6**ab**). However, both of these studies included affected individuals with extreme outlier profile changes,

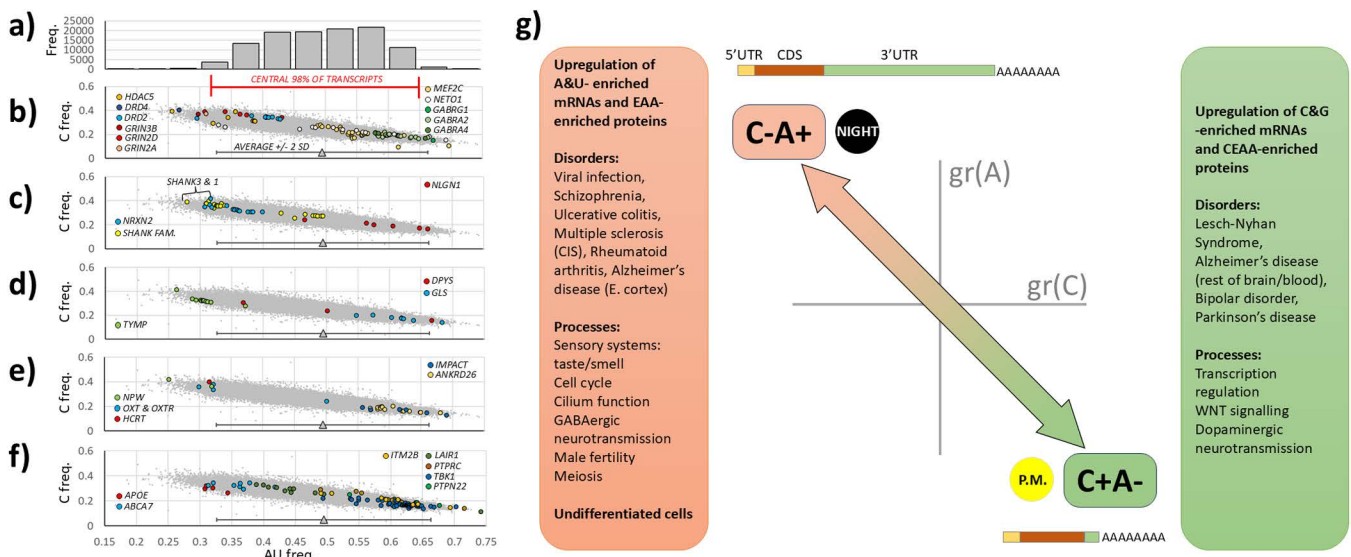

**Fig 7. The physiological and pathological roles of mRNAs at the extremes of composition.** *The extremes of composition are illustrated by the histogram of transcripts (a) within A+U frequency categories shown under (f) (for example, 0.45 indicates number of transcripts with A+U composition ≤ 0.45). The central 98% of all transcripts (red bar) and +/- 2 standard deviation bars (grey, b-f) illustrate the extreme composition of transcripts that fall outside of their limits. Extreme gene transcripts with key roles in brain function and disease (particularly neurodevelopmental disorders such as schizophrenia and autism spectrum disorder) are shown in (b) and (c). (d) contains nucleobase metabolism genes that may help cells reverse shifts in global expression. In (e), extreme transcripts dictating behavioural control of feeding are shown to occupy low A+U frequency extremes while the other end contains transcripts involved in cellular responses to nutritional deficit. (f) shows major risk genes for Alzheimer's disease and transcripts that may contribute to the immune/viral phenotypes associated with extreme C+A- and C-A+ shifts, respectively. A summary of the consequences of directional shifts in global expression profile, either C+A- or C-A+, are indicated in (g). These shifts occur over the circadian day ('night'/'p.m.'). They are also associated with distict average transcript sizes and subregion proportions between the 500 most A+U rich mRNAs (2,891 nucleotides, top), and the 500 most C+G rich (1,016 nucleotides, bottom). The text boxes indicate the major outcomes of C+A- and C-A+ shifts on encoded protein composition, association with disorders, and over-represented gene ontology terms at those extremes.*

suggesting a subgroup of cases with significant nucleobase effects. Such an ASD subgroup with nucleobase metabolism defects has been documented previously [19–21]. In the parietal cortex and frontal cortex of those diagnosed with MS, a similar direction-of-effect was observed, but this was not statistically significant compared to healthy controls (GSE131281, Fig 6**ac**).

Finally, a range of muscular/neuromuscular/neurodegenerative conditions were examined in small-scale studies GSE3307 (Fig 6**ad**) and GSE26927 (Fig 6**ae/af**). Amyotrophic lateral sclerosis (ALS/MND) and acute quadriplegic myopathy (AQM) showed significant C-A+ shifts in global profile of muscle biopsies, while Becker muscular dystrophy (BMD) and limb girdle muscular dystrophy (LGMD) showed significant changes in the opposite direction. The ALS findings were not replicated in spinal cord samples, but PD was associated with a significant C+A- shift in substantia nigra tissue gene expression profile, the anatomical origin of its neurodegenerative pathology.

### 'Hardwired' consequences of global transcriptional change

Although tissues, drugs, and diseases have been described here in terms of their impact on global expression, ultimately it is expression change at the *individual* gene/protein level that produces functional consequences. It was hypothesised that specific mRNA transcripts might have evolved compositional extremes so that cells could initiate a 'hardwired' expression response to any fluctuations in nucleobase availability. Expression changes in the proteins encoded by such extreme mRNAs might act to restore supplies of nucleobases or may offer survival/selective advantages in adverse metabolic conditions. Such a system would manifest as enriched/over-represented gene ontology terms associated with those mRNAs at compositional extremes. The 110,962 human GENCODE v44 protein-coding transcripts were ranked by single nucleobase composition, or A+U or C+G paired nucleobase composition. Those transcripts found at the most extreme 1% ends of each ranking were assessed using ENRICHR, GOnet, and WEB Gestalt online gene ontology and functionality resources [22–24]. Many statistically significant gene ontology enrichments were identified including C- or (C+G)-rich transcripts over-represented in terms Regulation of DNA-templated Transcription (GO:0006355, adjusted p-values $1.3 \times 10^{-12}$ and $1.3 \times 10^{-5}$, respectively), and DNA-binding transcription factor activity, RNA polymerase II-specific (GO:0000981, FDR adjusted p-values 0.0050 and $1.7 \times 10^{-6}$, respectively). Individually, C-richness was associated with Biological Process Involved in Intraspecies Interaction Between Organisms (GO:0051703, FDR adjusted p-value 0.0054) and G-richness with Regulation of Cellular Component Organization (GO:0051128, FDR adjusted p-value $1.8 \times 10^{-3}$). It would be predicted that expression of genes with these GO terms would be diminished as global profiles shift in the C-A+ direction.

At the other extreme, (A+U)-rich transcripts were over-represented in Meiotic Cell Cycle (GO:0051321, FDR adjusted p-value 0.014) and Synaptonemal Complex Organization (GO:0070193, FDR adjusted p-value 0.015). Single nucleobase analysis of U-rich transcripts identified gene ontology terms Nervous System Process (GO:0050877, FDR adjusted p-value 0) and Detection of Chemical Stimulus (GO:0009593, FDR adjusted p-value $6.3 \times 10^{-28}$) because of large clusters of olfactory and gustatory receptors at this extreme. A-rich transcripts, by contrast, were associated with Cilium Assembly (GO:0060271, FDR adjusted p-value $4.5 \times 10^{-5}$), Cell Cycle (GO:0007049, FDR adjusted p-value $1.5 \times 10^{-9}$), and Nuclear Chromosome Segregation (GO:0098813, FDR adjusted p-value $4.2 \times 10^{-8}$). Genes with these GO terms would have reduced expression when global profiles shift in the C+A- direction.

Together, these data suggest that any biochemical stressor that compromises nucleobase availability will potentially reduce metabolic expenditure by affecting transcription rates of mRNAs/proteins promoting cellular transcription or proliferation, while simultaneously engaging behaviours that increase survival through greater resource-seeking or restricted resource allocation. This narrative closely resembles our previous analysis of protein function at the extremes of nutritional amino acid composition [1].

Figs 7a-f examine the identities and functionalities of mRNAs located at the compositional extremes, paying particular regard to known genetic risk factors for those common disorders discussed in Fig 6. A frequency distribution of A+U

composition in the entire human transcriptome is shown in Fig 7a. Aligned directly underneath are A+U vs. C composition scatterplots for all transcripts in grey, with the mean and +/- 2 standard deviations of A+U composition indicated (Fig 7b). The central 98% of transcripts are also indicated by the red bracket.

The brain appears to use extreme composition to regulate development and activity (Figs 7b & 7c). The top 1% (C+G)-rich (=A+U-poor) transcripts include neurotransmitter receptors GRIN3B, GRIN2D, and GRIN2A (NMDA excitatory action, glutamatergic), DRD2/4 (dopamine), and CHRNA4 (acetylcholine); acetylcholinesterase, ACHE; neuronal channels KCNQ2 and CACNG6; synaptic structural proteins SHANK1&3; neuroligin 2 post-synaptic protein, NLGN2 (often associated with inhibitory neurons), and its pre-synaptic interactor neurexin 2, NRXN2. At the other extreme, (A+U)-rich transcripts include the neurotransmitter receptors GABRA2, GABRA4, and GABRG1 (inhibitory action, GABAergic); channels SCN2A, KCNT2, CACNA2D and CACNB4; neuroligin 1, NLGN1 (often associated with excitatory neurons), and its interactor, neurexin 1, NRXN1. HDAC5 is at the opposite extreme to the MEF2C gene it suppresses. Together with extreme transcript, NETO1 (a GRIN2A interactor), these three proteins are known to modulate synaptic plasticity. This catalogue of opposing distributions of opposing functionalities suggests that the brain can leverage nucleobase-mediated shifts in global transcriptional profile to control the tone of neuronal and circuit activity (excitatory vs. inhibitory) while also modulating trans-synaptic interactions. It also renders these genes and their functional systems susceptible to dysregulation by pathological changes in global profile – many have established genetic links to neuropsychiatric and neurodevelopmental disorders [25–27].

Multiple nucleobase metabolism enzymes, such as TYMP and DPYS, are encode by transcripts located at compositional extremes (Fig 7d), potentially offering the hardwired feedback regulation by nucleobase availability suggested earlier. Two (A+U)-rich transcripts, GLS (glutaminase) and ASNS (asparagine synthetase), encode the enzymes that catabolise L-glutamine, the single amino acid starting-point for purine and pyrimidine base synthesis. A cell with reduced glutamine resources would have limited nucleobase synthesis, potentially lowering GLS/ASNS transcription levels and therefore preventing further glutamine loss through catabolism. Such a homeostatic mechanism has received experimental support from metabolomic analysis of GLS hypermorphs in which nucleobase synthesis is impaired [28]. Transcripts linking nucleobase deficits with homeostatic feeding behaviour outputs are found in the (C+G)-rich extremity (Fig 7e): NPW (feeding and drinking behaviours), oxytocin (OXT) and its receptor (OXTR)(social/offspring bonding, reproduction, and food consumption), and hypocretin (HCRT/orexin, arousal, hunger). At the A+U extreme, IMPACT and ANKRD26 are two genes that mediate cellular adjustments to nutritional deficiency. Collectively, these nutritional findings directly complement our published protein analysis [1] in which EAA-rich proteins within the leptin signalling pathway provided a novel mechanistic connection between amino acid deficiency and hunger response.

Established individual neurodevelopmental, neurodegenerative, and autoimmune disease susceptibility genes such as DRD2, APOE, ABCA7, TGFB1, TERT, POMC, HDAC5, WNT1, CARD9, LRP5, NRXN2, SHANK3&1, HBA, TH, PRRT2 and IGF2 are rich in C+G, whereas SNCA, PTPN22, LRRK2, TLR1, and PTH are rich in A+U nucleobases. Their susceptibility to global expression change may help explain the disease findings presented in Fig 6 and may even contribute to the nosological classification of these disorders. For example, Alzheimer's disease susceptibility genes ABCA7 and APOE are strongly enriched in C+G nucleobases, as shown in Fig 7f. At the other extreme, out of the total of 110,962 transcripts examined, one transcript of LAIR1 was ranked 2nd most A+U enriched, in close proximity to transcripts for PTPRC ('CD45', 6th), TBK1 (12th), and PTPN22 (111th). These four genes encode key components of the immunological and antiviral responses. Their expression is likely to be increased following the C-A+ expression changes associated with viral infection and autoimmune status, as set out in Fig 6. Returning to Alzheimer's disease, the transcript ranked 26th most A+U enriched encodes the ITM2B (BRI2) protein, an inhibitor of amyloid precursor protein processing by secretases [29].

## Discussion

Re-examination of transcriptomic datasets reveals that drug treatment and disease pathology can cause coordinated distortions of cellular gene expression that are independent of promoter activity. Multiple lines of evidence are presented that

these distortions result from interactions of nucleobase availability and mRNA nucleobase composition. As such, mRNA transcription joins DNA replication [30,31] and protein translation [1] to complete the trio of large-scale cellular polymerisation reactions for which substrate concentration is a regulatory influence.

This system of global regulation can be usefully compared to the principles of market economics. Each mRNA sequence defines the *demand*, the nucleobase availability is the *supply*, and the *price* is the biosynthetic cost of each transcript (assumed to be inversely proportional to its expression). Nucleobase 'supply shock' resulting from drug or disease action alters the 'market equilibrium' by changing the cost of biosynthesis of each mRNA, ultimately guiding each transcript to a new equilibrium of expression level. The global profile, therefore, can be thought of as the totality of all expressed transcript equilibria.

The unchanging composition of mRNA transcripts results in an unchanging profile of genes that are up- or down-regulated according to the directionality (C+A-/C-A+) and scale of the global profile shift, irrespective of drug or disease identity. The origin of the nucleobase supply deficits inferred here is simple when considering nucleobase analogues, antimetabolites, or inborn errors of metabolism such as Lesch-Nyhan syndrome. However, the majority of drugs and diseases perturbing global expression that are listed here do not easily fit this model, possessing a diverse range of well-characterised pharmacological actions or pathological underpinnings that do not currently invoke nucleobase supply issues. It is, therefore, speculated that they must all, at some level, compromise the metabolic capabilities and outputs of the cell, leading to an indirect impact on purine and pyrimidine biochemistries. For example, the alkylating agents have a very specific effect on DNA replication that does not appear to regulate nucleobase supply directly. However, it may be the case that these agents have non-DNA targets such as enzymes (cysteine alkylation), or that their detoxification depletes glutathione reserves, causing downstream metabolic stress. Targeted examination of the correlations between metabolomic and transcriptomic datasets will be required to test this concept of generalised metabolic disruption leading to global expression profile change.

An alternative explanation to the proposed supply/demand model arises when it is considered that mRNA base composition and transcript length are covariates. This rival mechanistic model would state that transcript length (for example, through drug/disease effects on transcriptional processivity or transcript half-life) is the primary determinant of the observed global gene expression effects, although this stance cannot easily explain analogue effects or the asymmetry of some drug profiles.

The observed global changes in expression profile were largely limited to two directions, here termed C+A- and C-A+. Five consequences of this constrained directionality are immediately apparent.

Firstly, the precise phenotypic outcomes will vary from one organ to another as each tissue's unique fingerprint of actively expressed genes presents a different substrate for the profile-defining conditions to act on. This may explain why diseases with an involvement of global expression profile dysregulation have tissue-specific pathologies.

Secondly, the bidirectional mechanism offers a more nuanced interpretation of conventional models of genetic susceptibility to disease or drug response because risk-carrying variants located within genes at the extremes of composition (for example, *SHANK3* for autism spectrum disorder or *APOE* or *ABCA7* for Alzheimer's disease) will be subject to reduced or increased impact (expressivity/penetrance) according to the direction of profile perturbation. As such, global expression control is a potential channel for environmental influence on genetic risk in instances where GxE effects operate.

Thirdly, the bidirectional effect offers a unifying model to explain the observed similarities and co-morbidities for certain disorders: for example, the established links between schizophrenia (C-A+) and autoimmune conditions (C-A+) [32], and how viral infection (C-A+) can be a trigger for both [33,34]. These may indicate synergistic interactions due to a shared direction of action. Conditions such as epilepsy, that share many features with other neuropsychiatric disorders and possess risk genes with extreme composition, should be considered prime candidates for future study on this basis.

Fourthly, a drug acting to drive the global expression profile in one direction (for example, methotrexate, C+A-) might derive its beneficial effects as a 'DMARD' by reversing an opposite expression profile shift caused by a pathological

condition (for example, rheumatoid arthritis, C-A+). Employing the same logic, cancer chemotherapy agents such as methotrexate, sulindac, cytarabine, trichostatin A, cyclophosphamide, and azaribine detailed in Fig 4 seem to defy simple treatment categories – possessing antiviral, anti-inflammatory, and epigenetic actions. It is tempting to speculate that these shared actions are all a manifestation of their directional impacts on perturbed global transcriptomes.

Lastly, directional effects are experienced to various degrees by all transcripts across the composition spectrum, producing the breadth of expression changes that result in the observed global profile change. Such wide-ranging changes result in phenotypic *effects* (disease presentation/drug effect/drug adverse effect) but are likely to confound attempts to determine root *causes* of disease or drug action through transcriptomics. A relevant example of this problem is the pervasive transcriptome-wide gene expression changes associated with *MYC* gene deregulation. These have defied explanation through straightforward models of transcription factor action [35,36] but MYC's established role in nucleotide synthesis regulation [37] might suggest global expression perturbation as an alternative explanation. MYC's additional (or potentially entirely linked) roles as a driver of pluripotency and cancer also match observations here of a highly skewed baseline transcriptional profile in transformed and undifferentiated cell types such as ES cells.

The range of chemicals listed in Fig 4 offer new perspectives on drug action and adverse effects. An acknowledged limitation of this work is that the original choice of the agents was subjective (often a focus on toxicity) and the effects of individual drugs are often restricted to a single study, preventing a broader comparisons of drug effects between tissues/cell types. However, the 12 antibiotics were an unexpected discovery given their traditionally accepted modes of action. Beta-lactams were found to act in all directions, but the remaining antibiotic classes only move profiles in a C-A+ direction. A recent multi-omic study of pathogen response to antibiotics identified purine/pyrimidine metabolism perturbation as a major target [38]. If this effect is shared with eukaryotic cells *in vivo*, antibiotic effectiveness or side-effects might be associated with global changes to both host and pathogen transcriptomes. Supporting this, accumulating evidence indicates that use of supplemental antibiotics in cell culture media can directly affect eukaryotic cell behaviour [39].

More generally, side-effects and disease risks are reported for many of the prescription drugs listed in Fig 4. The idiosyncratic and shared nature of adverse effects (bone marrow suppression, developmental/teratogenic changes, ototoxicity, fatigue/sleep disturbance, xerostomia, keratoconjunctivitis sicca, headache, fertility, and Stevens-Johnson syndrome) when compared to the huge chemical diversity of drugs is a paradox that may find resolution through the shared dysregulation of the pathways and processes found at the extremes of base composition. For example, it could be hypothesised that transcripts of deafness-associated genes, *MSRB3*, *ESPN*, and *WHRN*, all in possession of extreme A+U/C+G compositions, are likely to be sensitive to any drug-mediated global dysregulation of expression, with ototoxicity a potential outcome. An example of a long-term drug risk, which has received recent attention, is the purported link between the use of proton pump inhibitors such as lansoprazole and pantoprazole (both present in Fig 4) and an increased risk of Alzheimer's disease [40], a condition that is shown here to be strongly associated with shifts in global transcriptional profile. If drug side-effects are indeed linked to global gene expression regulation, new small molecules may benefit from gene expression profile analysis to detect changes that could predict off-target risks.

The tight correlation between extreme A+U:C+G compositions within the 5'UTR, CDS, and 3'UTR of mRNAs and the nutritional amino acid composition of the resulting proteins was entirely unexpected because the allocation of codons to amino acids is considered an ancient 'frozen accident' that occurred millions of years before amino acids became categorisable by their nutritional source [41]. However, it might be postulated that a group of nine amino acids with codons highly biased in A+U nucleobase composition represented a 'fault line' in the genetic code that the animal kingdom took advantage of by outsourcing their metabolically expensive biosynthesis to organisms occupying lower trophic levels – those amino acids becoming 'essential' from that point forward. This seemingly irrational biological step was perhaps made feasible because global expression regulation might reduce the 'ribotoxic' consequences of EAA dietary shortfall

 

on stalled protein production [42]. Specifically, instances of poor nutrition, and the resulting compromised metabolic state impacting nucleobase supply, could act as a 'canary in the coal mine' that would pre-emptively reduce transcription of extreme A+U composition mRNAs encoding those proteins particularly rich in the nine EAAs. Indeed, it could be further speculated that UTR effects on protein composition indicate that natural selection drove UTR sequence mutation to match the CDS nucleobase composition to maximise the sensitivity of the 'antenna-like' detection of nucleobase supply deficits by transcripts, thus enhancing limitations on the unsupportable translation of extreme EAA proteins. mRNAs that are part of important biological processes could have subsequently co-opted A+U/C+G composition to control their own regulation. Both this nutritional connection and the previously described drug/toxin effects provide a clear and quantifiable route for GxE interactions that are an important feature of complex phenotypes and illness.

In summary, it is now important to determine the place of global expression regulation within the hierarchy of disease aetiopathology – how it is connected to the root cause of disease and where it sits in relation to the other global pathologies such as epigenetic dysregulation, oxidative stress, ER stress, and chronic inflammation. Similarly, a vital future step is determining whether therapeutic manipulation of the global expression profile is a viable approach to treat disease.

## Methods

All analysis, calculations, graphing, and statistics were carried out in Microsoft Excel.

110,962 human mRNA transcripts (the protein-coding component of the GENCODE v44) were downloaded in FASTA format from (https://www.gencodegenes.org/). These represent the known splice variants of the ~20,000 protein-coding genes. Download of mRATBN7.2 (NCBI) was used for the rat transcript analysis. Download of Gencode.vM33 was used for the mouse transcript analysis. Nucleobase frequencies and lengths were calculated for each individual transcript using the same Excel formula approach detailed in [1]. Combination frequencies were also calculated, for example, A+U. Thus, every transcript had nucleobase compositional data that could be correlated with expression data from chip-based microarray studies.

Microarray-based gene expression datasets were downloaded from Gene Expression Omnibus (GEO, hosted by the National Center for Biotechnology Information, NCBI) [43]. All deposited data within GEO adhere to the ethical NCBI/GEO Human Subject Guidelines, where required. No human gene expression data can be traced back to an individual. GEO datasets were accessed between November 2023 and October 2024. =VLOOKUP was used to link microarray chip identifiers associated with gene expression data to the correct transcript from the nucleobase composition analysis, thus permitting correlation. To examine the composition effect of a specific nucleobase on expression, the spreadsheet was sorted by one of C/A/G/U/AU/CG/AG/CU frequencies. Expression data for each sample/experiment across transcripts (typically deposited in $\log_2$ transformed form) were divided into 25 equal-sized bins from the lowest to the highest frequency of the searched nucleobase composition. Within each bin, expression values were averaged. To calculate profile gradients, gr(X), for each sample or experiment, the inverse log of the 2nd bin was subtracted from the inverse log of the 24th bin and then the result divided by the change in nucleobase composition between the corresponding bins. The extreme (1st/25th) bins were not used because they often showed greater variance. Corrected gradients cgr(X) were calculated by subtracting experimental control gradients from experimental sample gradients. For the screen, 'outlier' drugs listed in Fig 4 and S2 Table deviated from the mean of the controls by greater than 1 standard deviation of the control values in at least two of the four nucleobase dimensions.

## Supporting information

**S1 Table. The Gene Expression Omnibus accession codes for all microarray gene expression studies re-analysed in this manuscript are detailed within this table.** The associated publications are referenced and information on the breakdown of sample numbers in case:control studies provided.
(DOCX)

**S2 Table. Thirteen datasets were analysed to identify drugs/chemical entities with pharmacological actions on global gene expression.** In this table the full corrected gradient data (cgr(X)) are provided for all studies, together with colour highlighting to indicate drugs producing gradients of profile that substantially deviated from the control condition averages by at least 1 standard deviation ('sd control'). Abbreviations: CMC, Carboxymethyl cellulose thickening agent; SDS, sodium dodecyl sulphate; F/M, Female and Male CD-1 mice (in GSE44783).
(XLSX)

**S1 Figure. Further investigation of the effects of nucleobase supply and mRNA composition demand using drug treatment data.** In (a) a reconstruction of Fig 3 is shown to draw attention to the relative global expression profile effect changes resulting from treatment with three drugs, bleomycin A2, urethane, and mitomycin C. The chemical structures of these three drugs are shown to highlight their different direct pharmacological actions. In (b) the relative proportions of two of the four nucleobases (C and A) have been estimated within the total expressed mRNA population (assumed to be in equilibrium with free pools of cellular nucleobases) by multiplying each transcript's expression level by its base composition frequency and then summing all products. For each drug treatment, the specific nucleobase investigated was expressed as a proportion of the total of all four sums. Histogram values are the averages, and the error bars are the 95% confidence intervals, of nucleobase proportions after replicated drug treatments. T-tests were used to determine if nucleobase proportion averages (C – blue, A – orange) were significantly different between treatments or the water controls. In (c) the identities of the top 100 upregulated transcripts for each of the three drug treatments were compared for overlaps and the results visualised in the Venn diagram inset. The expression levels of the top 100 most upregulated bleomycin A2 transcripts were compared between bleomycin A2 and urethane (red), and bleomycin A2 and mitomycin c (green). Linear trendlines and $R^2$ values are shown for both correlations.
(TIF)

**S2 Figure. Transcriptomic studies of specific brain tissues in which a diagnosis of Alzheimer's disease did not result in a significant change to global expression profile.** Study GSE36980 (a) comprised 15 frontal cortex samples from individuals diagnosed with Alzheimer's disease (AD FC), 18 frontal cortex samples from healthy controls (HC FC), 10 temporal cortex samples from individuals diagnosed with Alzheimer's disease (AD TC), 19 temporal cortex samples from healthy controls (HC TC), 8 hippocampal samples from individuals diagnosed with Alzheimer's disease (AD HIP), and 10 hippocampal samples from healthy controls (HC HIP). Study GSE118553 (b,c) comprised 52 temporal cortex samples from individuals diagnosed with Alzheimer's disease (Temporal C AD av), 32 temporal cortex samples from individuals who were asymptomatic (intact cognition but pathology consistent with AD, Temporal C Asympt av), 31 temporal cortex samples from healthy control individuals (Temporal C Ctl av), 40 frontal cortex samples from individuals diagnosed with Alzheimer's disease (Frontal C AD av), 33 frontal cortex samples from individuals who were asymptomatic (Frontal C Asympt av), and 23 frontal cortex samples from healthy control individuals (Frontal C Ctl av).Symbols represent average gr(C) and gr(A) values for a tissue/diagnosis and are plotted on the same type of scatterplots as Figure 5, with error bars corresponding to 95% confidence intervals.
(TIFF)

## Author contributions

**Conceptualization:** Benjamin S. Pickard.

**Data curation:** Benjamin S. Pickard.

**Formal analysis:** Benjamin S. Pickard.

**Investigation:** Benjamin S. Pickard.

**Methodology:** Benjamin S. Pickard.

**Validation:** Benjamin S. Pickard.

**Visualization:** Benjamin S. Pickard.

**Writing – original draft:** Benjamin S. Pickard.

**Writing – review & editing:** Benjamin S. Pickard.

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
