## [Decision Letter · Decision Letter 0]

10 Jan 2025

PONE-D-24-58748Gene expression is globally regulated by interacting nucleobase supply and mRNA composition demand: a mechanism disrupted by multiple disease states and drug treatmentsPLOS ONE

Dear Dr. Pickard,

Thank you for submitting your manuscript to PLOS ONE. After careful consideration, we feel that it has merit but does not fully meet PLOS ONE’s publication criteria as it currently stands. Therefore, we invite you to submit a revised version of the manuscript that addresses the points raised during the review process.

 Based on the reviews received by the referees, it would seem that several issues must be addressed before this manuscript could be considered worthy of acceptance. All referees were unanimous in their belief that the figures were of significantly low resolution, impeding their ability to fully comprehend them. Beyond this, three other referees expressed some issues with certain details in the manuscript. Having looked at these comments, I feel that for the most part, they are indeed valid questions that must be taken into account in a revision. Given the significance of this work, which I believe will be of great benefit to those working in this and related fields, I look forward to seeing a revised version that addresses each of these comments.

We look forward to receiving your revised manuscript.

Kind regards,

Robert John O'Reilly, M.D., Ph.D.

Academic Editor

PLOS ONE

Journal Requirements:

Reviewers' comments:

Reviewer's Responses to Questions

**Comments to the Author**

1. Is the manuscript technically sound, and do the data support the conclusions?

Reviewer #1: Yes

Reviewer #2: Yes

Reviewer #3: No

Reviewer #4: Partly

2. Has the statistical analysis been performed appropriately and rigorously? 

Reviewer #1: Yes

Reviewer #2: No

Reviewer #3: I Don't Know

Reviewer #4: N/A

3. Have the authors made all data underlying the findings in their manuscript fully available?

Reviewer #1: Yes

Reviewer #2: Yes

Reviewer #3: Yes

Reviewer #4: Yes

4. Is the manuscript presented in an intelligible fashion and written in standard English?

Reviewer #1: Yes

Reviewer #2: Yes

Reviewer #3: No

Reviewer #4: Yes

5. Review Comments to the Author

Reviewer #1: The most important aspect of this paper is linking metabolism to expression directly. It is a new perspective that should be taken into account in future attempts to develop novel drugs. In my opinion, the authors analysis makes a strong case for this proposed idea. The paper is well-written and has the potential to contribute significantly to the fields of drug development and sheds light on how viral infections and other diseases can influence expression. The author is advised to attempt to increase the resolution of the images as some were difficult to see.

Reviewer #2: The author conducted a study that introduces a novel approach to evaluating the impact of nucleobase composition on the expression levels of mRNA sequences. This approach involves re-analyzing 45 publicly available transcriptomic datasets. This paper might be published after the authors address some concerns, as follows:

1) The current title is too dense and combines multiple concepts, making it difficult to understand. It is recommended to revise the title by simplifying the language, which will help a broader audience grasp the key message of the research while preserving its essential information.

2) Section titles are too long

3) The quality of the figures is too low, making it difficult to read the labels. As a result, this reviewer is unable to properly assess the discussion or compare it with the figures.

4) Although the limitations of the study are presented especially on the agents used, the authors need to include any potential biases in the dataset selection or analysis methods.

5) Expand on the mechanistic insights provided by the study, particularly how nucleobase supply and mRNA composition interact at a molecular level.

6) The author needs to expand the conclusion.

7) Did the author do data scaling? It would be good to normalize or standardize the gene expression data to ensure comparable scales across conditions or genes. For example, z-score transformation can be useful for ensuring comparability. A principal component analysis could also reduce the dimensionality of the data while preserving variance, allowing you to visualize patterns in gene expression across conditions. A PCA plot can summarize the main sources of variation and how different samples or conditions cluster.

8) Another confirmatory tool would be the use of statistical tests such as ANOVA, t-tests, or non-parametric tests (e.g., Kruskal-Wallis) to assess the significance of observed differences across conditions. Present p-values or confidence intervals alongside radar plots to support the interpretations.

Reviewer #3: The author needs to describe their methods, and the design better. The nucleotide composition of any given transcripts are affected by many factors including alternative splicing. It is not clear how the author define it. It is also impossible to estimate the nucleotide composition by computing the GC/AT content in transctiptome data. It's even not clear if nucleotide avaliability is a limiting factor for gene expression in most context.

All the data are presented with bad quality and can not provide any information

Reviewer #4: Thank you for the opportunity to review this paper, which reports on the potential role of nucleobase supply in determining mRNA expression levels. This is a reasonable proposition, as potential restrictions in this supply would reasonably be expected to force the cell to prioritise the mRNA complement, and by extension, the protein profile of the cell. The strength of this paper lies in the amount of work that has been done to demonstrate a correlation between the types of mRNA transcripts across cell types and the prevalence of certain bases within those transcripts. Unfortunately, in places, the text reads as if the data is being constrained to justify the hypothesis, rather than the conclusions arising directly from the data. Additionally, some of the conclusions seem to stretch a bit too far, particularly considering the absence of direct measures of nucleobase availability and promotor activity. There are other aspects of this piece of scholarship that would benefit from clarification, additional detail, and/or revision. Some specific comments are provided in the attached document.

6. PLOS authors have the option to publish the peer review history of their article (what does this mean? ). If published, this will include your full peer review and any attached files.

**Do you want your identity to be public for this peer review?** For information about this choice, including consent withdrawal, please see our Privacy Policy .

Reviewer #1: No

Reviewer #2: No

Reviewer #3: No

Reviewer #4: No

---

## [Author Response · Author response to Decision Letter 1]

31 Jan 2025

PONE-D-24-58748 author response to reviewer comments

The author would like to express his sincere thanks to the four reviewers for their extensive, thoughtful and helpful comments. Below, the author has set out how each of their points has been addressed, individually or collectively, in the revised manuscript. All line numbers relate to the ‘track changes, show all changes’ version.

1) Firstly, apologies to all reviewers for the poor image quality in the original submission, which, I imagine, made the reviewing process much harder than it needed to be. All problem figures have been re-scaled. The versions visible in the assembled PDF are still not of suitable resolution, but by clicking on the links in the top right of the page the high-resolution versions should download for review.

2) ‘Data not shown’ has been removed from the MS text (line 477) and, instead, a new supplemental figure 2 has been created.

3) R#2: The main title and section titles have been substantially reworked and simplified.

4) R#4: The abstract has been substantially reworked to remove reference to Lesch-Nyhan Syndrome (Line 33) which the author agrees was perhaps over-emphasised. Related, R#4: ‘it is recommended that the author reconsider the order in which the data is presented, as this would strengthen the correlations being drawn between the data from the HPRT knockdown cells and certain drugs with the findings in Figures 1-3’. I believe this comment has been addressed by the de-emphasising of Lesch-Nyhan/HPRT in the text and re-wording.

5) R#4: ‘what is meant by “…simple numerical outputs”? In the context of an Abstract, this comes across as a bit vague and uninformative’. RESPONSE: The abstract text has been changed to provide more explicit detail of the gradient calculation (Line 23).

6) R#4: The reviewer sought clarification of the concepts of supply of nucleobases/amino acids and how this impacts global expression through mRNA or protein sequence composition. The Introduction section now has additional text starting on Line 67 and 83 which hopefully clarifies the logic for the study – which is based on mRNA composition analysis rather than direct nucleobase quantification. The issue of lack of direct quantification of nucleobases is also addressed in further points below.

7) R#3: ‘the authors need to include any potential biases in the dataset selection or analysis methods’, ‘Expand on the mechanistic insights provided by the study, particularly how nucleobase supply and mRNA composition interact at a molecular level.’, and

‘The author needs to expand the conclusion’. R#4:’ it would be helpful for the author to explain how the frequency of individual bases within the transcripts is transformed to demonstrate unequivocally that the promoters do not contribute to altered mRNA levels’. ‘drugs that are alkylating agents will change the availability of certain transcripts due to the DNA damage caused, but how does this correlate with the availability of nucleobases themselves?’,’ Figure 4, which should provide evidence in this regard, is incredibly difficult to read but fluorouracil, for example, had an asymmetrical effect, the meaning of which is left unexplained, and would be expected to skew the transcript profile’. RESPONSE: These comments have been grouped together as they all focus on measurement and interpretation of nucleobase supply. They have prompted a much more extensive Discussion section, starting on Line 682, that examines potential confounders, mechanistic explanations, and the issues of very defined drug actions that do not appear to affect nucleobases. Additionally, point 10, below, details further support for the nucleobase supply hypothesis.

8) R#3: ‘Did the author do data scaling? It would be good to normalize or standardize the gene expression data to ensure comparable scales across conditions or genes. For example, z-score transformation can be useful for ensuring comparability. A principal component analysis could also reduce the dimensionality of the data while preserving variance, allowing you to visualize patterns in gene expression across conditions. A PCA plot can summarize the main sources of variation and how different samples or conditions cluster. Another confirmatory tool would be the use of statistical tests such as ANOVA, t-tests, or non-parametric tests (e.g., Kruskal-Wallis) to assess the significance of observed differences across conditions. Present p-values or confidence intervals alongside radar plots to support the interpretations.’.R#3: ‘The author needs to describe their methods, and the design better. The nucleotide composition of any given transcripts are affected by many factors including alternative splicing. It is not clear how the author define it. It is also impossible to estimate the nucleotide composition by computing the GC/AT content in transcriptome data.’ R#4: ‘what constitutes an “outlier” is unclear’. RESPONSE: I have addressed these comments together because they largely relate to Methods. Over 100,000 human transcripts were studied derived from transcription of the ~20K protein-coding genes. Thus, multiple splicing is taken into account when calculating compositional information. Equally importantly, most microarray chips used in the generation of the data have the capability to distinguish a proportion of the splice variation in the transcriptome. Thus, the potential action of alternative splicing on the analysis is already accounted for, as much as the data allow. Where mRNA-encoded protein correlations were made, the average mRNA nucleobase composition of all gene X spliceforms was used to compare to protein X amino acid composition. In terms of scaling, the use of gradients or control-corrected gradients is stated in the paper as a means to generate outputs that permit some limited comparison between studies. I did attempt to apply some form of scaling between the 13 studies examining drug effects on global expression profile. This was not successful, and I explain that drug potency on global expression seems to be very tissue-specific (Line 368 and onwards), further convincing me that quantitative cross-study/cross-tissue comparisons are not likely to be possible or biologically useful. Thus, nothing other than direction of effect is shown when outlier drugs are collated in Figure 4. The Supplemental Data 2 for this section does allow others to examine absolute drug potency within studies. On the topic of ‘outlier’ definition, this is visible within Supplemental Data 2, and is already formally defined in the Methods (enhancements made to text description), and the legends to Figs. 3 and 4 (again, text added). The statistical analysis requested is present in Figure 6 in the form of 95% confidence interval limits, and the labelled instances where two-tailed T-tests indicated statistically significant profile differences between conditions. Again, I must apologise for the resolution of the images obscuring this valuable information. Overall, I have responded to these comments by adding greater detail to the Methods section of the paper.

9) R#4: ‘were the bins independent of the tissue type is the early studies?’ and ‘there appears to be a contradiction between the information in the Methods section, lines 716-717, which states that transcripts, “…were divided into 25 equal-sized bins from the lowest to highest frequency…” and the comment in line 86 that there were “…25 equal-sized bins each containing hundreds of randomly allocated transcripts”’. RESPONSE: Bins only relate to transcript composition and not to tissue origin. They are used to visually simplify the complexity of global expression and to provide a means to calculate the ‘gradient’ of the global profile. The actual analyses of nucleobase effects on expression are always conducted on bins of compositionally sorted transcripts (lowest frequency of base X in bin 1 all the way to highest frequency of base X in bin 25). However, when this concept is first introduced on Line 96, a ‘sham’ analysis (Fig.1b) without ranking was applied such that when compositional ranking was actually introduced (Fig.1c and Fig.1d), the impact on the gradient becomes immediately apparent. I have re-phrased the analytical description in this section to remove any confusion and to explicitly label and highlight the ‘sham’ comparison.

10) R#4: ‘since there aren’t any direct measures of free nucleobase quantities or promotor activity in this section, it would be helpful for the author to explain how the frequency of individual bases within the transcripts is transformed to demonstrate unequivocally that the promoters do not contribute to altered mRNA levels’, ‘Repeatedly across Figures 1-3, the author draws conclusions about the nucleobase supply, correlating it with nucleobase composition in the absence of evidence that the supply is perturbed. I have chosen not to enumerate all of the statements of concern, as this would have run to many pages, but certain key issues are identified below.’. ‘the strongest data in this study comes from the HPRT knockdown cells (lines 373-377). That said, correlation does not equal causation, and therefore, the fact that the C+A- global transcriptome change seen in these cells resembles that seen in other data does not mean that they have the same root cause.’ And ‘the first sentence of the Discussion draws conclusions about promoter activity when no evidence has been provided to demonstrate this and the second sentence reinforces the tenuous nature of the conclusions drawn, using “appear to result” (line 586), which can be taken to emphasise the possible correlation in the absence of evidence of causation’ RESPONSE: The reviewer indicates a number of statements and claims throughout the manuscript which perhaps rely too heavily on the initial assumptions of a nucleobase supply model. I have collated many of these comments in this response point. In addition to new text in the Introduction that acknowledges the lack of direct nucleobase quantification (Line 67 and, separately, 83), I have added two new analyses to accompany the data in Figure 3 (from Line 290, and supplemental Figure 1). They cannot replace direct quantification of nucleobases as a gold standard, but I think they convincingly support the hypothesised mechanism and therefore justify claims made elsewhere. As a basis for the first analysis, I claim (and this is implicit in the model) that there will be an equilibrium between the free nucleobase pool and the pool of nucleobases incorporated into the total cellular mRNA. This allows the inference of nucleobase supply change after drug treatment by determining if the relative abundance of each of the nucleobases locked up in cellular transcripts has changed (an actual measure that is possible to make from the expression data). This is shown to be the case. Secondly, it is determined that two very different drugs (structure/pharmacological action) can be selected - but with similar gradient values and direction - and shown to have almost identical influences on global expression change (gene identity and expression level correlation). I believe these two new observations cannot be reconciled with a promoter-based explanation and, instead, support a ‘universal’ influence on gene expression based on changing nucleobase supply coupled with unchanging mRNA composition. I thank the reviewer for demanding better supportive evidence for the theoretical claims.

11) R#4: ‘it is unclear how coding DNA sequences that result in proteins rich in essential amino acids provides any insight into nucleobase availability (lines183-187) or why this study was needed to illustrate a skew in the nucleobases required for the different amino acids. Simply counting the different bases in the genetic code across the types of amino acids would have demonstrated this without the need for experimentation and therefore, the comment that it was “unexpected” seems unusual. I have obviously missed the point and therefore, would find it helpful for the author to provide additional information to explain their meaning.’ And ‘ untranslated regions are known to regulate expression (e.g. 5’UTR regulates ribosome recruitment, translation efficiency, etc.) and therefore, protein availability, so how does the nucleobase composition of 5’UTR or 3’UTR provide insights into nucleobase availability (lines 202-205)?’. RESPONSE: The ‘prequel’ paper to this one (Thompson & Pickard, 2024) identified two of the three nutritionally classified amino acid groupings (essential, conditionally essential amino acids) as constraints on protein expression, much as A+U, and C+G nucleobases are identified as constraints on mRNA expression in this manuscript. I wished to test whether there was any formal relationship between these two very distinct forms of building-block supply constraint on expression level. Remarkably, A+U enriched (=A+U supply susceptible) transcripts preferentially encode proteins enriched in essential amino acids, whereas C+G enriched transcripts encode conditionally essential amino acid enriched proteins. That is unexpected because it is a bit like arbitrarily dividing a room full of many people into two groups and finding one group owns cats and the other dogs. It seems incredibly unlikely. However, this nucleobase-amino acid alignment may offer a clue to its origin/purpose: any metabolic shortfall in A+U has ‘reach-through’ to reduce the synthesis of essential amino rich proteins. I proposed in the text that the risks of starvation conditions (in which there is no essential amino acid intake) include causing significant cellular stress, as ribosomes try but fail to make proteins with many essential amino acids in them. This stress might be mitigated, in part, by those same starvation conditions affecting nucleobase metabolism, reducing A+U availability, and therefore the reducing transcription of mRNAs likely to code for those ‘dangerous’ essential amino acid-rich proteins. In fact, I went one stage further by claiming that, while the mutational loss of biochemical machinery for synthesising essential amino acids in higher organisms might be of energetic benefit, it could only have happened in the context of the safeguarding offered by global mRNA expression control. One more aspect is discussed (and raised as an issue by R#4): the observation that the 5’UTR or 3’UTR non-coding sequences of a transcript also influence amino acid type in their encoded proteins. While it’s intuitive that coding regions offer reach-through to protein in the global expression model, it seems to be the case that it has been selectively beneficial for UTR sequences to evolve a shared similarity in nucleobase composition to their adjoining ORFs because this would increase the sensitivity of the entire transcript to a nucleobase supply-mediated form of expression regulation. I believe this speculation is warranted as it provides a testable hypothesis for the origin and original purpose of global expression control and the emergence of the animal kingdom (and also suggests yet another function for UTRs). It also links diet (=environment) with the disease pathologies described (=genes), which I believe is a useful route into the underdeveloped gene x environment (GxE) model of disease susceptibility. Starting at Line 191 and Line 727 additional text and a relevant, supportive reference has been added.

12) R#4: regarding the number of replicate treatments (line 241), what does “approximately twelve replicate treatments” mean? RESPONSE: The legend has been altered for clarity: “Data points represent the average of the biological replicates of expression quantification for each drug, with replicates typically numbering twelve.”

13) R#4: limitations of this work (lines 221-223) belong in the Discussion, not the Results. RESPONSE: text moved to Discussion

14) R#4: ‘the figure titles are descriptive sentences rather than titles and therefore, it is suggested that they be revised’. RESPONSE: legend titles have a been shortened and altered to indicate both method and principal outcome.

15) R#4: ‘“asymmetrical” (“Assymm.”) should be defined’ RESPONSE: Text on Line 239 has been expanded

16) R#4: ‘clarification in the figure legend of the x-axis values in F

---

## [Decision Letter · Decision Letter 1]

26 Feb 2025

A mechanism of global gene expression regulation is disrupted by multiple disease states and drug treatments

PONE-D-24-58748R1

Dear Dr. Pickard,

We’re pleased to inform you that your manuscript has been judged scientifically suitable for publication and will be formally accepted for publication once it meets all outstanding technical requirements.

Kind regards,

Robert John O'Reilly, M.D., Ph.D.

Academic Editor

PLOS ONE

Additional Editor Comments (optional):

Reviewers' comments:

Reviewer's Responses to Questions

**Comments to the Author**

1. If the authors have adequately addressed your comments raised in a previous round of review and you feel that this manuscript is now acceptable for publication, you may indicate that here to bypass the “Comments to the Author” section, enter your conflict of interest statement in the “Confidential to Editor” section, and submit your "Accept" recommendation.

Reviewer #2: All comments have been addressed

2. Is the manuscript technically sound, and do the data support the conclusions?

Reviewer #2: Yes

3. Has the statistical analysis been performed appropriately and rigorously? 

Reviewer #2: Yes

4. Have the authors made all data underlying the findings in their manuscript fully available?

Reviewer #2: Yes

5. Is the manuscript presented in an intelligible fashion and written in standard English?

Reviewer #2: Yes

6. Review Comments to the Author

Reviewer #2: The authors have effectively addressed all the comments and suggestions provided by the reviewers, and the manuscript is now deemed suitable for publication. However, a minor suggestion regarding the supplementary materials: it would be beneficial to combine all Tables and Figures into a single file to make it easier for readers to download the supplementary content.

7. PLOS authors have the option to publish the peer review history of their article (what does this mean? ). If published, this will include your full peer review and any attached files.

**Do you want your identity to be public for this peer review?** For information about this choice, including consent withdrawal, please see our Privacy Policy .

Reviewer #2: No

---

## [Editor Report · Acceptance letter]

PONE-D-24-58748R1

PLOS ONE

Dear Dr. Pickard,

I'm pleased to inform you that your manuscript has been deemed suitable for publication in PLOS ONE. Congratulations! Your manuscript is now being handed over to our production team.

Kind regards,

on behalf of

Dr. Robert John O'Reilly

Academic Editor

PLOS ONE